# PointNeXt: Revisiting PointNet++ with Improved Training and Scaling Strategies

**Guocheng Qian**[1*], **Yuchen Li**[1*], **Houwen Peng**[2†],
**Jinjie Mai**[1], **Hasan Abed Al Kader Hammoud**[1], **Mohamed Elhoseiny**[1], **Bernard Ghanem**[1†]
[1]King Abdullah University of Science and Technology (KAUST), [2]Microsoft Research

## Abstract

PointNet++ is one of the most influential neural architectures for point cloud understanding. Although the accuracy of PointNet++ has been largely surpassed by recent networks such as PointMLP and Point Transformer, we find that a large portion of the performance gain is due to improved training strategies, *i.e.* data augmentation and optimization techniques, and increased model sizes rather than architectural innovations. Thus, the full potential of PointNet++ has yet to be explored. In this work, we revisit the classical PointNet++ through a systematic study of model training and scaling strategies, and offer two major contributions. First, we propose a set of improved training strategies that significantly improve PointNet++ performance. For example, we show that, without any change in architecture, the overall accuracy (OA) of PointNet++ on ScanObjectNN object classification can be raised from $77.9\%$ to $86.1\%$, even outperforming state-of-the-art PointMLP. Second, we introduce an inverted residual bottleneck design and separable MLPs into PointNet++ to enable efficient and effective model scaling and propose *PointNeXt*, the next version of PointNets. PointNeXt can be flexibly scaled up and outperforms state-of-the-art methods on both 3D classification and segmentation tasks. For classification, PointNeXt reaches an overall accuracy of $87.7\%$ on ScanObjectNN, surpassing PointMLP by $2.3\%$, while being $10\times$ faster in inference. For semantic segmentation, PointNeXt establishes a new state-of-the-art performance with $74.9\%$ mean IoU on S3DIS (6-fold cross-validation), being superior to the recent Point Transformer. The code and models are available at https://github.com/guochengqian/pointnext.

## 1 Introduction

Recent advances in 3D data acquisition have led to a surge in interest for point cloud understanding. With the rise of PointNet [27] and PointNet++ [28], processing point clouds in their unstructured format using deep CNNs become possible. Subsequent to "PointNets", many point-based networks are introduced with the majority focusing on developing new and sophisticated modules to extract local structures, *e.g.* the pseudo-grid convolution in KPConv [41] and the self-attention layer in Point Transformer [54]. These newly proposed methods outperform PointNet++ by a large margin in a variety of tasks, leaving the impression that the PointNet++ architecture is too simple to learn complex point cloud representations. In this work, we revisit PointNet++, the classical and widely used network, and find that its full potential has yet to be explored, mainly due to two factors that were not present at the time of PointNet++: (1) superior training strategies and (2) effective model scaling strategies.

Through a comprehensive empirical study on various benchmarks, *e.g.*, ScanObjecNN [42] for object classification and S3DIS [1] for semantic segmentation, we discover that training strategies, *i.e.*, data augmentation and optimization techniques, play an important role in the network's performance. In fact, a large part of the performance gain of state-of-the-art (SOTA) methods [44, 41, 54] over Point-Net++ [28] is due to improved training strategies that are, unfortunately, less publicized compared to

---

*Equal contribution. †Corresponding authors.

36th Conference on Neural Information Processing Systems (NeurIPS 2022).

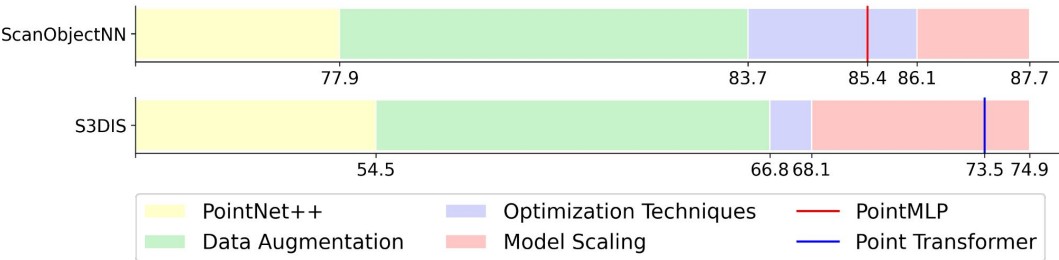

Figure 1: **Effects of training strategies and model scaling on PointNet++** [28]. We show that improved training strategies (data augmentation and optimization techniques) and model scaling can significantly boost PointNet++ performance. The average overall accuracy and mIoU (6-fold cross-validation) are reported on ScanObjectNN [42] and S3DIS [1].

architectural changes. For example, randomly dropping colors during training can unexpectedly boost the testing performance of PointNet++ by 5.9% mean IoU (mIoU) on S3DIS [1], as demonstrated in Tab. 5. In addition, adopting label smoothing [37] can improve the overall accuracy (OA) on ScanObjectNN [42] by 1.3%. These findings inspire us to revisit PointNet++ and equip it with new advanced training strategies that are widely used today. Surprisingly, as shown in Fig. 1, utilizing the improved training strategies alone improves the OA of PointNet++ by 8.2% on ScanObjectNN (from 77.9% to 86.1%), establishing a new SOTA without introducing any changes to the architecture (refer to Sec. 4.4.1 for details). For the S3DIS segmentation benchmark, the mIoU evaluated in all areas by 6-fold cross-validation can increase by 13.6% (from 54.5% to 68.1%), outperforming many modern architectures that are subsequent to PointNet++, such as PointCNN [21] and DeepGCN [20].

Moreover, we observe that the current prevailing models [19, 41, 54] for point cloud analysis have employed many more parameters than the original PointNets [27, 28]. Effectively expanding PointNet++ from its original small scale to a larger scale is a topic worth studying because larger models are generally expected to enable richer representations and perform better [2, 18, 53]. However, we find that the naive way of using more building blocks or increasing the channel size in PointNet++ only leads to an overhead in latency and no significant improvement in accuracy (see Sec. 4.4.2). For effective and efficient model scaling, we introduce residual connections [12], an inverted bottleneck design [34], and separable MLPs [30] into PointNet++. The modernized architecture is named PointNeXt, the next version of PointNets. PointNeXt can be scaled up flexibly and outperforms SOTA on various benchmarks. As demonstrated in Fig. 1, PointNeXt improves the original PointNet++ by 20.4% mIoU (from 54.5% to 74.9%) on *S3DIS* [1] 6-fold and achieves 9.8% OA gains on *ScanObjecNN* [42], surpassing SOTA Point Transformer [54] and PointMLP [26]. We summarize our contributions next:

- We present the first systematic study of training strategies in the point cloud domain and show that *PointNet++ strikes back* (+8.2% OA on ScanObjectNN and +13.6% mIoU on S3DIS) by simply adopting *improved training strategies alone*. The improved training strategies are general and can be easily applied to improve other methods [27, 44, 26].

- We propose PointNeXt, the next version of PointNets. PointNeXt is scalable and surpasses SOTA on all tasks studied, including object classification [42, 47], semantic segmentation [1, 5], and part segmentation [51], while being faster than SOTA in inference.

## 2    Preliminary: A Review of PointNet++

Our PointNeXt is built upon PointNet++ [28], which uses a U-Net [33] like architecture with an encoder and a decoder, as visualized in Figure 2. The encoder part hierarchically abstracts features of point clouds using a number of *set abstraction* (SA) blocks, while the decoder gradually interpolates the abstracted features by the same number of *feature propagation* blocks. The SA block consists of a *subsampling* layer to downsample the incoming points, a *grouping* layer to query neighbors for each point, a set of shared multilayer perceptrons (*MLPs*) to extract features, and a *reduction* layer to aggregate features within the neighbors. The combination of the grouping layer, MLPs, and the reduction layer is formulated as:

$$\mathbf{x}_i^{l+1} = \mathcal{R}_{j:(i,j)\in\mathcal{N}} \left\{ h_{\boldsymbol{\Theta}} \left( [\mathbf{x}_j^l; \mathbf{p}_j^l - \mathbf{p}_i^l] \right) \right\}, \tag{1}$$

where $\mathcal{R}$ is the reduction layer (*e.g.* max-pooling) that aggregates features for point $i$ from its neighbors denoted as $\{j : (i, j) \in \mathcal{N}\}$. $\mathbf{p}_i^l, \mathbf{x}_i^l, \mathbf{x}_j^l$ are the input coordinates, the input features, and the features of neighbor $j$ in the $l^{th}$ layer of the network, respectively. $h_\Theta$ denotes the shared MLPs that take the concatenation of $\mathbf{x}_j^l$ and the relative coordinates $(\mathbf{p}_j^l - \mathbf{p}_i^l)$ as input. Note that, since PointNet++ with single-scale grouping that uses one SA block per stage is the default architecture used in the original paper [28], we refer to it as PointNet++ throughout and use it as our baseline.

## 3   Methodology: From PointNet++ to PointNeXt

In this section, we present how to modernize the classical architecture PointNet++ [28] into PointNeXt, the next version of PointNet++ with SOTA performance. Our exploration mainly focuses on two aspects: (1) training modernization to improve data augmentation and optimization techniques, and (2) architectural modernization to probe receptive field scaling and model scaling. Both aspects have important impact on the model's performance, but were under-explored by previous studies.

### 3.1   Training Modernization: PointNet++ Strikes Back

We conduct a systematic study to quantify the effect of each data augmentation and optimization technique used by modern point cloud networks [44, 41, 54] and propose a set of improved training strategies. The potential of PointNet++ can be unveiled by adopting our proposed training strategies.

#### 3.1.1   Data Augmentation

Data augmentation is one of the most important strategies to boost the performance of a neural network; thus we start our modernization from there. The original PointNet++ used simple combinations of data augmentations from random rotation, scaling, translation, and jittering for various benchmarks [28]. Recent methods adopt stronger augmentations than those used in PointNet++. For example, KPConv [41] randomly drops colors during training, Point-BERT [52] uses a common point resampling strategy to randomly sample $1,024$ points from the original point cloud for data scaling, while RandLA-Net [14] and Point Transformer [54] load the entire scene as input in segmentation tasks. In this paper, we quantify the effect of each data augmentation through an additive study.

We start our study with PointNet++ [28] as the baseline, which is trained with the original data augmentations and optimization techniques. We remove each data augmentation to check whether it is necessary or not. We add back the useful augmentations but remove the unnecessary ones. We then systematically study all the data augmentations used in the representative works [44, 41, 30, 54, 26, 52], including data scaling such as point resampling [52] and loading the entire scene as input [14], random rotation, random scaling, translation to shift point clouds, jittering to add independent noise to each point, height appending [41] (*i.e.*, appending the measurement of each point along the gravity direction of objects as additional input features), color auto-contrast to automatically adjust color contrast [54], and color drop that randomly replaces colors with zero values. We verify the effectiveness of data augmentation incrementally and only keep the augmentations that give a better validation accuracy. At the end of this study, we provide a collection of data augmentations for each task that allow for the highest boost in the model's performance. Sec. 4.4.1 presents and analyzes in detail the uncovered findings.

#### 3.1.2   Optimization Techniques

Optimization techniques including loss functions, optimizers, learning rate schedulers, and hyperparameters are also vital to the performance of a neural network. PointNet++ uses the same optimization techniques throughout its experiments: CrossEntropy loss, Adam optimizer [15], exponential learning rate decay (Step Decay), and the same hyperparmeters. Owing to the development of machine learning theory, modern neural networks can be trained with theoretically better optimizers (*e.g.* AdamW [25] *vs.* Adam [15]) and more advanced loss functions (CrossEntropy with label smoothing [37]). Similarly to our study on data augmentations, we also quantify the effect of each modern optimization technique on PointNet++. We first perform a sequential hyperparameter search for the learning rate and weight decay. We then conduct an additive study on label smoothing, optimizer, and learning rate scheduler. We discover a set of improved optimization techniques that further

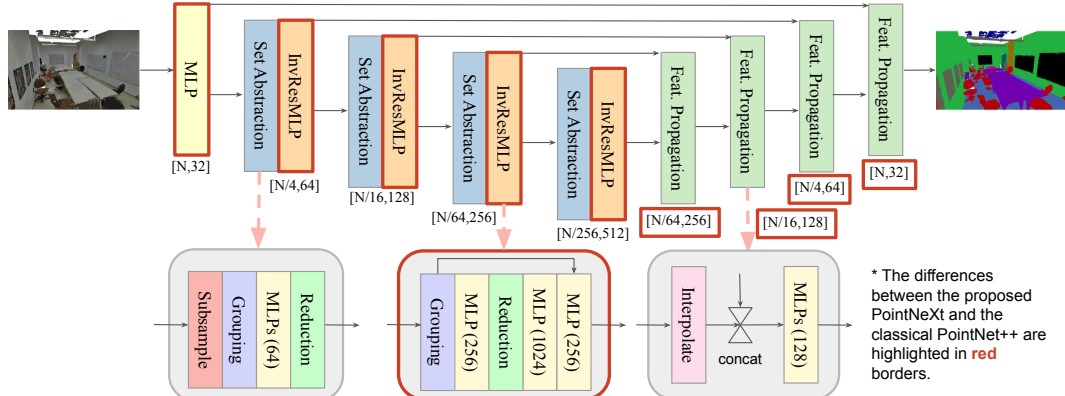

Figure 2: **PointNeXt architecture.** PointNeXt shares the same Set Abstraction and Feature Propagation blocks as PointNet++ [28], while adding an additional MLP layer at the beginning and scaling the architecture with the proposed Inverted Residual MLP (InvResMLP) blocks.

boost performance by a decent margin. In general, CrossEntropy with label smoothing, AdamW, and Cosine Decay can decently optimize models in various tasks. See Sec. 4.4.1 for detailed findings.

## 3.2 Architecture Modernization: Small Modifications → Big Improvements

In this subsection, we modernize PointNet++ [28] into the proposed PointNeXt. The modernization consists of two aspects: (1) receptive field scaling and (2) model scaling.

### 3.2.1 Receptive Field Scaling

The receptive field is a significant factor in the design space of a neural network [36, 6]. There are at least two ways to scale the receptive field in point cloud processing: (1) adopting a larger radius to query the neighborhood, and (2) adopting a hierarchical architecture. Since the hierarchical architecture has been adopted in the original PointNet++, we mainly study (1) in this subsection. Note that the radius of PointNet++ is set to an initial value $r$ that doubles when the point cloud is downsampled. We study a different initial value in each benchmark and discover that the radius is dataset-specific and can have significant influence on performance. This is elaborated in Sec. 4.4.2.

Furthermore, we find that the relative coordinates $\Delta_p = \mathbf{p}_j^l - \mathbf{p}_i^l$ in Eq. (1) make network optimization harder, leading to a decrease in performance. Thus, we propose relative position normalization ($\Delta_p$ normalization) to divide relative position by the neighborhood query radius:

$$\mathbf{x}_i^{l+1} = \mathcal{R}_{j:(i,j)\in\mathcal{N}} \left\{ h_{\Theta} \left( [\mathbf{x}_j^l; (\mathbf{p}_j^l - \mathbf{p}_i^l)/r^l] \right) \right\}. \tag{2}$$

Without normalization, values of relative positions ($\Delta_p = \mathbf{p}_j^l - \mathbf{p}_i^l$) are considerably small (less than the radius), requiring the network to learn a larger weight to apply on $\Delta_p$. This makes the optimization non-trivial, especially since weight decay is used to reduce the weights of the network and thus tends to ignore the effects of relative position. The proposed normalization alleviates this issue by rescaling and in the meantime reduces the variance of $\Delta_p$ among different stages.

### 3.2.2 Model Scaling

PointNet++ is a relatively small network, where the encoder consists of only 2 stages in the classification architecture and 4 stages for segmentation. Each stage consists of only 1 SA block, and each block contains 3 layers of MLP. The model sizes of PointNet++ for both classification and segmentation are less than 2M, which is much smaller compared to modern networks that typically use more than 10M parameters [41, 26, 30]. Interestingly, we find that neither appending more SA blocks nor using more channels leads to a noticeable improvement in accuracy, while causing a significant drop in throughput (refer to Sec. 4.4.2), mainly due to vanishing gradient and overfitting. Therefore, in this subsection, we study how to scale up PointNet++ in an effective and efficient way.

We propose an Inverted Residual MLP (InvResMLP) block to be appended after the first SA block, per stage, for effective and efficient model scaling. InvResMLP is built on the SA block and is

illustrated at the bottom middle of Fig. 2. There are three differences between InvResMLP and SA. (1) A residual connection between the input and the output is added to alleviate the vanishing gradient problem [12], especially when the network goes deeper. (2) Separable MLPs are introduced to reduce computation and reinforce pointwise feature extraction. While all 3 layers of MLPs in the original SA block are computed on the neighborhood features, InvResMLP separates the MLPs into a single layer computed on the neighborhood features (between the grouping and reduction layers) and two layers for point features (after reduction), as inspired by MobileNet [13] and ASSANet [30]. (3) The inverted bottleneck design [34] is leveraged to expand the output channels of the second MLP by 4 times to enrich feature extraction. Appending InvResMLP blocks is proven to significantly improve performance compared to the appending of the original SA blocks (see Sec. 4.4.2).

In addition to InvResMLP, we present three changes in the macro architecture. (1) We unify the design of PointNet++ encoder for classification and segmentation, *i.e.*, scaling the number of SA blocks for classification from 2 to 4 while keeping the original number (4 blocks) for segmentation at each stage. (2) We utilize a symmetric decoder in which its channel size is changed to match the encoder. (3) We add a stem MLP, an additional MLP layer inserted at the beginning of the architecture, to map the input point cloud to a higher dimension.

In summary, we present PointNeXt, the next version of PointNets [27, 50], modified from PointNet++ by incorporating the proposed InvResMLP and the aforementioned macro-architectural changes. The architecture of PointNeXt is illustrated in Fig. 2. We denote the channel size of the stem MLP as $C$ and the number of InvResMLP blocks as $B$. A larger $C$ leads to an increase in the width of the network (*i.e.*, width scaling), while a larger $B$ leads to an increase in the depth of the network (*i.e.*, depth scaling). Note that when $B = 0$, only one SA block and no InvResMLP blocks are used at each stage. The number of MLP layers in the SA block is set to 2, and a residual connection is added inside each SA block. When $B \neq 0$, InvResMLP blocks are appended after the original SA block. The number of MLP layers in the SA block in this case is set to 1 to save computation cost. The configuration of our PointNeXt family is summarized as follows:

- PointNeXt-S: $C = 32, B = 0$
- PointNeXt-B: $C = 32, B = (1, 2, 1, 1)$
- PointNeXt-L: $C = 32, B = (2, 4, 2, 2)$
- PointNeXt-XL: $C = 64, B = (3, 6, 3, 3)$

## 4 Experiments

We evaluate PointNeXt on five standard benchmarks: *S3DIS* [1] and *ScanNet* [5] for semantic segmentation, *ScanObjectNN* [42] and *ModelNet40* [47] for object classification, and *ShapeNetPart* [3] for object part segmentation.

**Experimental Setups.** We train PointNeXt using CrossEntropy loss with label smoothing [37], AdamW optimizer [25], an initial learning rate $lr = 0.001$, weight decay $10^{-4}$, with Cosine Decay, and a batch size of 32, with a 32G V100 GPU, for all tasks, unless otherwise specified. The best model on the validation set is selected for testing. For S3DIS segmentation, point clouds are voxel downsampled with a voxel size of 0.04m following common practice [41, 30, 54]. PointNeXt is trained with an initial $lr = 0.01$, for 100 epochs (training set is repeated by 30 times), using a fixed number of points $(24, 000)$ per batch with a batch size of 8 as input. During training, the input points are obtained by querying the nearest neighbors of a random point in each iteration. Following Point Transformer [54], we evaluate PointNeXt using the entire voxel-downsampled scene as input. For ScanNet scene segmentation, we follow the Stratified Transformer [16] and train PointNeXt with multi-step learning rate decay and decay at [70,90] epochs with a decay rate of 0.1 without label smoothing. The voxel size is set to 0.02m and input number of points in training is set to $64, 000$. We train the model for 100 epochs (training set is repeated for 6 times) with a batch size of 2 per GPU with 8 GPUs. For ScanObjectNN classification, PointNeXt is trained with a weight decay of 0.05 for 250 epochs. Following Point-BERT [52], the number of input points is set to $1, 024$, where the points are randomly sampled during training and uniformly sampled during testing (denoted as point resampled augmentation). For ModelNet40 classification, PointNeXt is trained similarly as ScanObjectNN but for 600 epochs. For ShapeNetPart part segmentation, we train PointNeXt using a batch size of 8 per GPU with 4 GPUs, and Poly FocalLoss [17] as criterion, for 400 epochs. Following PointNet++, 2,048 randomly sampled points with normals are used as input for training and testing. The details of data augmentations used in S3DIS, ScanNet, ScanObjectNN, ModelNet40 and ShapeNetPart are detailed in Sec. 4.4.1.

Table 1: **3D semantic segmentation in S3DIS (evaluation by 6-Fold or in Area 5) and ScanNet V2.** For PointNeXt in S3DIS Area 5, the average results without voting in three random runs are reported. The improvements of PointNeXt over the original performance reported by PointNet++ [28] are highlighted in green color. PointNet++ (ours) denotes PointNet++ trained using our improved data augmentation and optmization techniques. Methods are in chronological order.

| Method | S3DIS 6-Fold | | S3DIS Area-5 | | ScanNet V2 | | Params. | FLOPs | Throughput |
|---|---|---|---|---|---|---|---|---|---|
| | mIoU (%) | OA (%) | mIoU (%) | OA (%) | Val mIoU (%) | Test mIoU (%) | M | G | (ins./sec.) |
| PointNet [27] | 47.6 | 78.5 | 41.1 | - | - | - | 3.6 | 35.5 | 162 |
| PointCNN [21] | 65.4 | 88.1 | 57.3 | 85.9 | - | 45.8 | 0.6 | - | - |
| DGCNN [44] | 56.1 | 84.1 | 47.9 | 83.6 | - | - | 1.3 | - | 8 |
| DeepGCN [20] | 60.0 | 85.9 | 52.5 | - | - | - | 3.6 | - | 3 |
| KPConv [41] | 70.6 | - | 67.1 | - | 69.2 | 68.6 | 15.0 | - | 30 |
| RandLA-Net [14] | 70.0 | 88.0 | - | - | - | 64.5 | 1.3 | 5.8 | 159 |
| BAAF-Net [31] | 72.2 | 88.9 | 65.4 | 88.9 | - | - | 5.0 | - | 10 |
| Point Transformer [54] | 73.5 | 90.2 | 70.4 | **90.8** | 70.6 | - | 7.8 | 5.6 | 34 |
| CBL [39] | 73.1 | 89.6 | 69.4 | 90.6 | - | 70.5 | 18.6 | - | - |
| PointNet++ [28] | 54.5 | 81.0 | 53.5 | 83.0 | 53.5 | 55.7 | 1.0 | 7.2 | 186 |
| PointNet++ (ours) | 68.1(+13.6) | 87.6(+6.2) | 63.2±0.4(+9.7) | 87.5±0.2(+4.5) | 57.2(+3.7) | - | 1.0 | 7.2 | 186 |
| **PointNeXt-S (ours)** | 68.0(+13.5) | 87.4(+6.4) | 63.4±0.8(+9.9) | 87.9±0.3(+4.9) | 64.5(+11.0) | - | 0.8 | 3.6 | **227** |
| **PointNeXt-B (ours)** | 71.5(+17.0) | 88.8(+7.8) | 67.3±0.2(+13.8) | 89.4±0.1(+6.4) | 68.4(+14.9) | - | 3.8 | 8.9 | 158 |
| **PointNeXt-L (ours)** | 73.9(+19.4) | 89.8(+8.8) | 69.0±0.5(+15.5) | 90.0±0.1(+7.0) | 69.4(+15.9) | - | 7.1 | 15.2 | 115 |
| **PointNeXt-XL (ours)** | **74.9** (+20.4) | **90.3** (+9.3) | **70.5**±0.3(+17.0) | 90.6±0.1(+7.6) | **71.5**(+18.0) | **71.2**(+15.5) | 41.6 | 84.8 | 46 |

For all experiments except ShapeNetPart segmentation, we do not conduct any voting [22]2, since it is more standard to compare the performance without using any ensemble methods as suggested by SimpleView [8]. However, we found that the performance in ShapeNetPart of nearly all models is quite close to each other, where it is hard to achieve state-of-the-art IoUs without voting. We also provide model parameters (Params.) and inference throughput (instances per second) for comparison. The throughput of all methods is measured using $128 \times 1024$ (batch size 128, number of points 1024) as input in ScanObjectNN and ModelNet40 and $64 \times 2048$ in ShapeNetPart. In S3DIS, $16 \times 15,000$ points are used to measure throughput following [30], since some methods [44, 19] could not process the whole scene due to memory constraints. The throughput of all methods is measured using an NVIDIA Tesla V100 32GB GPU and a 32 core Intel Xeon @ 2.80GHz CPU.

## 4.1  3D Semantic Segmentation in S3DIS and ScanNet

**S3DIS** [1] (Stanford Large-Scale 3D Indoor Spaces) is a challenging benchmark composed of 6 large-scale indoor areas, 271 rooms, and 13 semantic categories in total. The standard 6-fold cross-validation results in S3DIS are reported in Tab. 1. Note that the official PointNet++ [28] did not conduct experiments in S3DIS. Here, we use the results reported by PointCNN [21] for comparison. Our PointNeXt-S, the smallest variant, outperforms PointNet++ by 13.5%, 6.4%, and 10.2% in terms of mean IoU (mIoU), overall accuracy (OA), and mean accuracy (mAcc), respectively, while being faster in terms of throughput. The increased speed is due to the reduced number of layers in the SA block for PointNeXt-S (see Sec. 3.2.2). With the proposed model scaling, the performance of PointNeXt can be gradually boosted. For example, PointNeXt-L outperforms SOTA Point Transformer [54] by 0.4% in mIoU while being $3\times$ faster. Note that Point Transformer utilizes most of the improved training strategies of ours. PointNeXt-XL, the extra large variant, achieves mIoU/OA/mAcc of 74.9%/90.3%/83.0%, while running faster than Point Transformer. As a limitation, our PointNeXt-XL consists of more parameters and is more computationally expensive in terms of FLOPs, mainly due to channel expansion ($\times 4$) in the inverted bottleneck and doubled initial channel size ($C = 64$). We also provide the results of PointNeXt in S3DIS area 5 in the Tab. 1 with mean±std in three random runs, where PointNeXt achieves similar improvements as the 6-fold experiments.

**ScanNet** [5], another well-known large-scale segmentation dataset, contains 3D indoor scenes of various rooms with 20 semantic categories. We follow the public training, validation, and test splits, with 1201, 312 and 100 scans, respectively. For PointNet++, we use the results reported from the Stratified Transformer [16] for comparison. As shown in Tab. 1, we improve PointNet++ from $53.5\%$ mIou to $57.2\%$ mIoU in the validation set by adopting the improved training strategies (detailed in supplementary material). PointNeXt-S further gains $+11.0$ in val mIoU over the original PointNet++ mostly due to the use of a smaller radius ($0.1m \rightarrow 0.05m$) and relative position normalization. The performance in ScanNet improves steadily with the increase in model sizes. Our largest variant,

---

2The voting strategy combines results by using randomly augmented points as input to enhance performance.

Table 2: **3D object classification in ScanObjectNN and ModelNet40.** Averaged results in three random runs using $1024$ points as input without normals and without voting are reported.

| Method | ScanObjectNN (PB_T50_RS) | | ModelNet40 | | Params. | FLOPs | Throughput |
| | OA (%) | mAcc (%) | OA (%) | mAcc (%) | M | G | (ins./sec.) |
| --- | --- | --- | --- | --- | --- | --- | --- |
| PointNet [27] | 68.2 | 63.4 | 89.2 | 86.2 | 3.5 | 0.9 | 4212 |
| PointCNN [21] | 78.5 | 75.1 | 92.2 | 88.1 | 0.6 | - | 44 |
| DGCNN [44] | 78.1 | 73.6 | 92.9 | 90.2 | 1.8 | 4.8 | 402 |
| DeepGCN [19] | - | - | 93.6 | 90.9 | 2.2 | 3.9 | 263 |
| KPConv [41] | - | - | 92.9 | - | 14.3 | - | - |
| ASSANet-L [30] | - | - | 92.9 | - | 118.4 | - | 153 |
| SimpleView [8] | 80.5±0.3 | - | 93.0±0.4 | 90.5±0.8 | 0.8 | - | - |
| MVTN [11] | 82.8 | - | 93.5 | 92.2 | 3.5 | 1.8 | 236 |
| Point Cloud Transformer [10] | - | - | 93.2 | - | 2.9 | 2.3 | - |
| CurveNet [48] | - | - | 93.8 | - | 2.0 | - | 22 |
| PointMLP [26] | 85.4±1.3 | 83.9±1.5 | **94.1** | **91.3** | 13.2 | 31.3 | 191 |
| PointNet++ [28] | 77.9 | 75.4 | 91.9 | - | 1.5 | 1.7 | 1872 |
| PointNet++ (ours) | 86.1±0.7(+8.2) | 84.2±0.9(+8.8) | 92.8±0.1(+0.9) | 89.9±0.8 | 1.5 | 1.7 | 1872 |
| **PointNeXt-S (ours)** | **87.7**±0.4(+9.8) | **85.8**±0.6(+10.4) | 93.2±0.1(+1.3) | 90.8±0.2 | 1.4 | 1.6 | 2040 |

PointNeXt-XL outperforms PointNet++ by $18.0\%$ mIoU in validation and achieves $71.2\%$ mIoU in testing, beating the recent methods Point Transformer [54] and CBL [39].

## 4.2 3D Object Classification in ScanObjectNN and ModelNet40

ScanObjectNN [42] contains about $15,000$ real scanned objects that are categorized into 15 classes with $2,902$ unique object instances. Due to occlusions and noise, ScanObjectNN poses significant challenges to existing point cloud analysis methods. Following PointMLP [26], we experiment on PB_T50_RS, the hardest and most commonly used variant of ScanObjectNN. As reported in Tab. 2, the proposed PointNeXt-S surpasses existing methods by non-trivial margins in terms of both OA and mAcc, while using much fewer model parameters and running much faster. Built upon PointNet++ [28], PointNeXt achieves significant improvements over the originally reported performance of PointNet++, *i.e.* +9.8% OA and +10.4% mACC. This demonstrates the efficacy of the proposed training and model scaling strategies. PointNeXt also outperforms SOTA PointMLP [26] (*i.e.* +2.3% OA, +1.9% mACC), while running $10\times$ faster. This shows that PointNeXt is a simple, yet effective, and efficient baseline. Note that we did not experiment with upscaled variants of PointNeXt on this benchmark, since we found that the performance had saturated using PointNeXt-S mostly due to the limited scale of the dataset.

ModelNet40 [47] was a commonly used 3D object classification dataset, which has $40$ object categories, each of which contains 100 unique CAD models. However, recent works [11, 26, 32] show an increasing interest in the real-world scanned dataset ScanObejectNN compared to this synthesized dataset. Following this trend, we mainly benchmarked PointNeXt in ScanObjectNN. Here, we also provide our results in ModelNet40. Tab. 2 shows that advanced training strategies improve PointNet++ from 91.9% OA to 92.8% OA without any architecture change. PointNeXt-S ($C = 32$) outperforms the original reported PointNet++ by $1.3\%$ OA, while being faster. Note that PointNeXt-S with a larger width $C = 64$ can achieve a higher overall accuracy (94.0%).

## 4.3 3D Object Part Segmentation in ShapeNetPart

ShapeNetPart [51] is a widely-used dataset for object-level part segmentation. It consists of $16,880$ models from 16 different shape categories, 2-6 parts for each category, and 50 part labels in total. As shown in Tab. 3, our PointNeXt-S with default width ($C = 32$) obtains a performance comparable

Table 3: **Part segmentation in ShapeNetPart.**

| Method | ins. mIoU | cls. mIoU | Params. | FLOPs | Throughput |
| --- | --- | --- | --- | --- | --- |
| PointNet [27] | 83.7 | 80.4 | 3.6 | 4.9 | **1184** |
| DGCNN [44] | 85.2 | 82.3 | 1.3 | 12.4 | 147 |
| KPConv [41] | 86.4 | 85.1 | - | - | 44 |
| CurveNet [48] | 86.8 | - | - | - | 97 |
| ASSANet-L [30] | 86.1 | - | - | - | 640 |
| Point Transformer [54] | 86.6 | 83.7 | 7.8 | - | 297 |
| PointMLP [26] | 86.1 | 84.6 | - | - | 270 |
| Stratifiedformer [16] | 86.6 | 85.1 | - | - | 398 |
| PointNet++ [28] | 85.1 | 81.9 | 1.0 | 4.9 | 708 |
| **PointNeXt-S** | 86.7±0.0(+1.6) | 84.4±0.2(+2.5) | 1.0 | 4.5 | 782 |
| **PointNeXt-S (C=64)** | 86.9±0.1(+1.8) | 84.8±0.5(+2.9) | 3.7 | 17.8 | 331 |
| **PointNeXt-S (C=160)** | **87.0**±0.1(+1.9) | **85.2**±0.1(+3.3) | 22.5 | 110.2 | 76 |

to that of the SOTA CurveNet [48] and outperforms a large number of representative networks, such as KPConv [41] and ASSANet [30] in terms of both instance mean IoU (ins. mIoU) and throughput. Due to the small scale of ShapeNetPart, the model would overfit after being depth scaled. However, we find by increasing the width from 32 to 64 instead, PointNeXt can outperform CurveNet, while being over $4\times$ faster. It is also worth highlighting that PointNeXt with an even larger width ($C = 160$) reaches 87.0% Ins. mIoU, whereas the performance of point-based methods has saturated below this value for years. We highlight that we used voting only in ShapeNetPart by averaging the results of 10 randomly scaled input point clouds, with scaling factors equal to [0.8,1.2]. Without voting, we notice a performance drop around 0.5 instance mIoU.

## 4.4 Ablation and Analysis

Tab. 4 and Tab. 5 present additive studies for the proposed training and scaling strategies in ScanObjectNN [42] and S3DIS [1], respectively. We adopt the original PointNet++ as the baseline. In ScanObjectNN, PointNet++ was trained by [42] with CrossEntropy loss, Adam optimizer, a learning rate 1e-3, a weight decay of 1e-4, a step decay of 0.7 for every 20 epochs, and a batch size of 16, for 250 epochs, while using random rotation and jittering as data augmentations. The official PointNet++ did not conduct experiments in S3DIS dataset. We refer to the widely used reimplementation [50], where PointNet++ was trained with the same settings as ScanObjectNN except that only random rotation was used as augmentation. Note that for all experiments, we train all our models for 250 epochs in ScanObjectNN and for 100 epochs in S3DIS.

### 4.4.1 Training Strategies

**Data augmentation** is the first aspect that we study to modernize PointNet++. We draw four conclusions based on observations in Tab. 4 and 5. (1) Data scaling improves performance for both classification and segmentation tasks. For example, point resampling is shown to boost the performance by 2.5% OA in ScanObjectNN. Taking the entire scene as input instead of using the block or sphere subsampled input as done in PointNet++ [28] and other previous works [41, 20, 30] improves the segmentation result by 1.1% mIoU. (2) Height appending improves performance, especially for object classification. Height appending makes the network aware of the actual size of the objects, thus leading to an increase in accuracy (+1.1% OA). (3) Color drop is a strong augmentation that significantly improves the performance of tasks where colors are available. Adopting color drop alone adds 5.9% mIoU in S3DIS area 5. We hypothesize that color drop forces the network to focus more on the geometric relationships between points, which in turn improves performance. (4) Larger models favor stronger data augmentation. Whereas random rotation drops the performance of PointNet++ by 0.3% mIoU in S3DIS ($2^{nd}$ row in Tab. 5 data augmentation part), it is shown to be beneficial for larger-scale models (*e.g.* raises 1.5% mIoU on PointNeXt-B). Another example in ScanObjectNN shows that the removal of random jittering also adds 1.1% OA. In general,

Table 4: Additive study of sequentially applying training and scaling strategies for classification on ScanObjectNN. We use light green, purple, yellow, and pink background colors to denote data augmentation, optimization techniques, receptive field scaling, and model scaling, respectively.

| Improvements | OA (%) | Δ |
|---|---|---|
| PointNet++ | 77.9 | – |
| + Point resampling | 81.4 ± 0.6 | +3.5 |
| − Jittering | 82.5 ± 0.4 | +1.1 |
| + Height appending | 83.6 ± 0.4 | +1.1 |
| + Random scaling | 83.7 ± 0.2 | +0.1 |
| + Label Smoothing | 85.0 ± 0.5 | +1.3 |
| + Adam → AdamW | 85.6 ± 0.1 | +0.6 |
| + AdamW → SGD | 84.8 ± 0.1 | -0.8 |
| + Step Decay → Cosine Decay | 86.1 ± 0.7 | +0.5 |
| + Radius 0.2 → 0.15 | 86.4 ± 0.3 | +0.3 |
| + Normalizing $\Delta_p$ (Eqn. (2)) | 86.7 ± 0.3 | +0.3 |
| + Scale up (PointNeXt-S) | 87.7 ± 0.4 | +1.0 |

Table 5: Additive study of sequentially applying training and scaling strategies for segmentation on S3DIS area 5. $+/-$ denote adopting/removing the strategy.

| Improvements | mIoU (%) | Δ |
|---|---|---|
| PointNet++ | 51.5 | - |
| + Entire scene as input | 52.6 ± 0.5 | +1.1 |
| − Rotation | 52.9 ± 0.6 | +0.3 |
| + Height appending | 53.4 ± 0.4 | +0.5 |
| + Color drop | 59.3 ± 0.7 | +5.9 |
| + Color auto-contrast | 61.0 ± 0.4 | +1.7 |
| + $lr = 0.001 \to 0.01$ | 61.5 ± 0.5 | +0.5 |
| + Label Smoothing | 61.9 ± 0.1 | +0.4 |
| + Adam → AdamW | 62.5 ± 0.6 | +0.6 |
| + AdamW → SGD | 59.4 ± 0.5 | -3.1 |
| + Step Decay → Cosine Decay | 63.2 ± 0.4 | +0.7 |
| + Normalize $\Delta_p$ | 63.6 ± 0.4 | +0.4 |
| + Scale down (PointNeXt-S) | 63.4 ± 0.8 | -0.2 |
| + Scale up (PointNeXt-B) | 65.8 ± 0.5 | +2.4 |
| + Rotation | 67.3 ± 0.2 | +1.5 |
| + Scale up (PointNeXt-L) | 69.0 ± 0.5 | +1.7 |
| + Scale up (PointNeXt-XL) | 70.5 ± 0.3 | +1.5 |

with the improved data augmentations, the OA of PointNet++ in ScanObjectNN and the mIoU in S3DIS area 5 are increased by 5.8% and 9.5%, respectively.

**Optimization techniques** involve loss functions, optimizers, learning rate schedulers, and hyperparameters. As shown in Tab. 4 and 5, Label Smoothing, AdamW [25] optimizer, and Cosine Decay consistently boost performance in both classification and segmentation tasks. This reveals that the more developed optimization methods such as label smoothing and AdamW are generally good for optimizing a neural network. Compared to Step Decay, Cosine Decay is also easier to tune (usually only the initial and minimum learning rates are required) and can achieve a performance similar to Step Decay. Regarding hyperparameters, using a learning rate greater than that used in PointNet++ improves the segmentation performance in S3DIS.

In general, our training strategies consisted of stronger data augmentation and modern optimization techniques can increase the performance of PointNet++ from 77.9% to 86.1% OA in ScanObjectNN dataset, impressively surpassing SOTA PointMLP by 0.7%. The mIoUs in S3DIS area 5 and S3DIS 6-fold (illustrated in Fig. 1) are boosted by 11.7 and 13.6 absolute percentage points, respectively. Our observations imply that *a significant portion of the performance gap between classical PointNet++ and SOTA is due to the training strategies.*

**Generalize to other networks.** Although the training strategies are proposed for PointNet++ [28], we find that they can be applied to other methods such as PointNet [27], DGCNN [44], and PointMLP [26], and also improve their performance. Such generalizability is validated in ScanObjectNN [42]. As shown in Tab. 6, the OA of the representative methods can all be improved when equipped with our training strategies.

Table 6: **The generalizability of improved training strategies.** OA on ScanObjectNN of networks trained with improved training strategies is reported.

| Method | ours | $\Delta$ |
|---|---|---|
| PointNet [27] | $74.4 \pm 0.9$ | **+6.2** |
| DGCNN [44] | $86.0 \pm 0.5$ | **+7.9** |
| PointMLP [26] | $87.1 \pm 0.7$ | **+1.7** |

### 4.4.2 Model Scaling

**Receptive field scaling** includes both radius scaling and normalizing $\Delta_p$ defined in Eqn. (2), which are also validated in Tab. 4 and 5. The radius is dataset specific, while down-scaling the radius from 0.2 to 0.15 improves 0.3% OA in ScanObjectNN, keeping the radius the same as 0.1 achieves the best performance in S3DIS. Regarding normalizing $\Delta_p$, it improves the performance in ScanObjectNN and S3DIS by 0.3 OA and 0.4 mIoU, respectively. Furthermore, in Tab. 7, we show that normalizing $\Delta_p$ has a larger impact (2.3 mIoU in S3DIS dataset) on the bigger model PointNext-XL.

**Model scaling** scales PointNet++ by the proposed InvResMLP and some macro-architectural changes (see Sec. 3.2.2). In Tab. 4, we show that PointNeXt-S using the stem MLP, the symmetric decoder, and the residual connection in the SA block improves 1.0% OA in ScanObjectNN. Performance in the large-scale S3DIS dataset can be further unveiled (from 63.8% to 70.5% mIoU) by up-scaling PointNeXt-S using more blocks of the proposed InvResMLP, as demonstrated in Tab. 5.

Furthermore, in Tab. 7,we ablate each component of the proposed InvResMLP block and different stage ratios in S3DIS area 5 using the best-performed model PointNeXt-XL as the baseline. As observed, each architectural change indeed contributes to increased performance. Among all changes, the residual connection is the most essential, without which the mIoU will drop from 70.5% to only 64.0%. The separable MLPs increase 3.9% mIoU while speeding up the network 3 times. Removing the inverted bottleneck from the baseline leads to a drop of 1.5% mIoU with less than a 1% gain in speed. Adding more blocks inside each stage after removing inverted bottleneck can improve its performance to $69.7 \pm 0.3$ but is still lower than the baseline. Another possibility is to use

Table 7: **Ablate architectural changes on S3DIS.** $-$ and TP denote removing from baseline and throughput.

| Ablate | mIoU | $\Delta$ | TP |
|---|---|---|---|
| baseline (PointNeXt-XL) | $70.5 \pm 0.3$ | - | 45 |
| $-$ normalizing $\Delta_p$ | $68.2 \pm 0.7$ | **-2.3** | 45 |
| $-$ residual connection | $64.0 \pm 1.0$ | **-6.5** | 45 |
| $-$ stem MLP | $70.1 \pm 0.4$ | **-0.4** | 46 |
| $-$ Separable MLPs | $66.6 \pm 0.8$ | **-3.9** | 15 |
| $-$ Inverted bottleneck | $69.0 \pm 0.4$ | **-1.5** | 48 |
| $-$ Inverted bottleneck | $69.7 \pm 0.3$ | **-0.8** | 43 |
| stage ratio $\rightarrow$ (1:1:1:1) | $69.8 \pm 0.6$ | **-0.7** | 52 |
| stage ratio $\rightarrow$ (2:1:1:1) | $69.4 \pm 0.4$ | **-1.1** | 41 |
| stage ratio $\rightarrow$ (1:1:2:1) | $69.9 \pm 0.6$ | **-0.6** | 47 |
| stage ratio $\rightarrow$ (1:1:1:2) | $69.5 \pm 0.4$ | **-1.0** | 48 |
| stage ratio $\rightarrow$ (1:3:1:1) | $70.1 \pm 0.4$ | **-0.4** | 39 |
| naive width scaling | $59.4 \pm 0.1$ | **-11.1** | 43 |
| naive depth scaling | $63.4 \pm 0.5$ | **-7.1** | 53 |
| naive compound scaling | $62.3 \pm 1.2$ | **-8.2** | 24 |

bottleneck design to shrink the channel size by 4 times in the middle of the module, and expand the network width or depth to achieve the same speed as the baseline. However, the best performance of bottleneck design only achieves 1.4% less mIoU compared to inverted bottleneck. Tab. 7 also shows the performance of naive width scaling that increases the width of PointNet++ from 32 to 256 to match the throughput of PointNeXt-XL, naive depth scaling to append more SA blocks in PointNet++ to obtain the same number of blocks of PointNext-XL whose $B = (3, 6, 3, 3)$, and naive compound scaling to double the width of the naive depth scaled model to the same width as PointNeXt-XL ($C = 64$). Our proposed model scaling strategy achieves much higher performance than these naive scaling strategies, while being much faster.

## 5   Related Work

*Point-based methods* process point clouds directly using their unstructured format compared to voxel-based methods [9, 4] and multi view-based methods [35, 11, 8]. PointNet [27], the pioneering work of point-based methods, proposes to model the permutation invariance of points with shared MLPs by restricting feature extraction to be pointwise. PointNet++ [28] is presented to improve PointNet by capturing local geometric structures. Currently, most point-based methods focus on the design of local modules. [44, 43, 29] rely on graph neural networks. [49, 21, 41, 40] project point clouds onto pseudo grids to allow for regular convolutions. [46, 22, 23] adaptively aggregate neighborhood features through weights determined by the local structure. In addition, very recent methods leverage Transformer-like networks [54, 16] to extract local information through self-attention. Our work does not follow this trend in local module design. In contrast, we shift our attention to another important but largely under-explored aspect, *i.e.*, the training and scaling strategies.

*Training strategies* are studied recently in [2, 45, 24] on image classification. In the point cloud domain, SimpleView [8] is the first work to show that training strategies have a large impact on the performance of a neural network. However, SimpleView simply adopts the same training strategies as DGCNN [44]. On the contrary, we conducted a systematic study to quantify the effect of *each* data augmentation and optimization technique, and propose a set of improved training strategies that boost the performance of PointNet++ [28] and other representative works [27, 44, 26].

*Model scaling* can significantly improve the performance of a network, as shown in pioneering works in various domains [38, 53, 20]. Compared to PointNet++ [28] that uses parameters less than 2M, most current prevailing networks consist of parameters greater than 10 M, such as KPConv [41] (15M) and PointMLP [26] (13M). In our work, we explore model scaling strategies that can scale up PointNet++ in an effective and efficient manner. We offer practical suggestions on scaling technologies that improve performance, namely using residual connections and an inverted bottleneck design, while maintaining throughput by using separable MLPs.

## 6   Conclusion and Discussion

In this paper, we demonstrate that with improved training and scaling strategies, the performance of PointNet++ can be increased to exceed the current state of the art. More specifically, we quantify the effect of each data augmentation and optimization technique that are widely used today, and propose a set of improved training strategies. These strategies can be easily applied to boost the performance of PointNet++ and other representative works. We also introduce the Inverted Residual MLP block into PointNet++ to develop PointNeXt. We demonstrate that PointNeXt has superior performance and scalability over PointNet++ on various benchmarks while maintaining high throughput. This work aims to guide researchers toward paying more attention to the effects of training and scaling strategies and motivate future work in this direction.

**Limitation.** Even though PointNeXt-XL is one of the largest models among all representative point-based networks, its number of parameters (44M) is still below that of small networks in image classification such as ConNeXt-S [24] (50M) and ViT-B [7] (87M), and is far from their large variants, including ConvNeXt-XL (350M) and ViT-L (305M). We do not push the model size further, mainly due to the smaller-scale nature of point cloud datasets. Moreover, our work is limited to existing modules since the focus is not on introducing new architectural changes.

**Acknowledgement** The authors would like to thank the reviewers of NeurIPS'22 for their constructive suggestions. This work was supported by the KAUST Office of Sponsored Research through the Visual Computing Center (VCC) funding, as well as, the SDAIA-KAUST Center of Excellence in Data Science and Artificial Intelligence (SDAIA-KAUST AI).

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
