# PointNeXt: Revisiting PointNet++ with Improved Training and Scaling Strategies

## — Supplementary Material —

In this appendix, we provide additional content to complement the main manuscript:

- Appendix A: A detailed description of Tab. 7.
- Appendix B: Comparisons of training strategies for prior representative works and PointNeXt.
- Appendix C: Qualitative comparisons on S3DIS and ShapeNetPart.
- Appendix D: The architecture of PointNeXt for classification.
- Appendix E: Societal impact.

## A   Detailed Description for Manuscript Tab. 7

Naive width scaling increases the channel size of PointNet++ from 32 to 256 to match the throughput of the baseline model, PointNeXt-XL. Naive depth scaling refers to appending more SA blocks ($B = (3, 6, 3, 3)$, the same as PointNext-XL) in PointNet++. Furthermore, naive compound scaling doubles the width of naive depth scaled model to the same as PointNeXt-XL ($C = 64$). Compared to the PointNet++ trained with improved training strategies (63.2% mIoU, 186 ins./sec.), naive depth scaling (63.4% mIoU, 53 ins. / sec.) and naive width scaling (59.4% mIoU, 43 ins./sec.) only lead to a large overhead in throughput with insignificant improvement in accuracy. In contrast, our proposed model scaling strategy achieves much higher performance than the naive scaling strategies while being much faster. This can be observed by comparing PointNeXt-XL (70.5% mIoU, 45 ins./sec.) to the naive compound scaled PointNet++ (62.3% mIoU, 24 ins./sec.).

## B   Training Strategies Comparison

In this section, we summarize the training strategies used in representative point-based methods such as DGCNN [8], KPConv [6], PointMLP [4], Point Transformer [10], Stratified Transformer [3], PointNet++ [5], and our PointNeXt on S3DIS [1] in Tab. I, on ScanObjectNN [7] in Tab. II, on ScanNet [2] in Tab. III, and on ShapeNetPart [9] in Tab. IV, respectively.

## C   Qualitative Results

We provide qualitative results of PointNeXt-XL for S3DIS (Fig. II) and PointNeXt-S ($C = 160$) for ShapeNetPart (Fig. III). The qualitative results of PointNet++ trained with the original training strategies are also included in the figures for comparison. On both datasets, PointNeXt produces predictions closer to the ground truth compared to PointNet++. More specifically, on S3DIS shown in (Fig. II), PointNeXt is able to segment hard classes, including doors ($1^{st}$, $3^{rd}$, and $4^{th}$ rows), clutter ($1^{st}$ and $3^{rd}$ rows), chairs ($2^{nd}$ row), and the board ($4^{th}$ row), while PointNet++ fails to segment properly to some extent. On ShapeNetPart (Fig. III), PointNeXt precisely segments wings of an airplane ($1^{st}$ row), microphone of an earphone($2^{nd}$ row), body of a motorbike($3^{rd}$ row), fin of a rocket($4^{th}$ row), and bearing of a skateboard ($5^{th}$ row).

## D   Classification Architecture

As illustrated in Fig. I, the classification architecture shares the same encoder as the segmentation one. The output features of the encoder are passed to a global pooling layer (*i.e.* global max-pooling) to acquire a global shape representation for classification. Note that the points are only downsampled by a factor of 2 in each stage, since the number of input points in classification tasks is usually small, *e.g.* 1024 or 2048 points.

Table I: **Training strategies used in different methods for S3DIS segmentation.**

| Method | DGCNN | KPConv | PointTransformer | PointNet++ | PointNeXt (Ours) |
|---|---|---|---|---|---|
| Epochs | 101 | 500 | 100 | 32 | 100 |
| Batch size | 12 | 10 | 16 | 16 | 8 |
| Optimizer | Adam | SGD | SGD | Adam | AdamW |
| LR | $1 \times 10^{-3}$ | $1 \times 10^{-2}$ | 0.5 | $1 \times 10^{-3}$ | 0.01 |
| LR decay | step | step | multi step | step | cosine |
| Weight decay | 0 | $10^{-3}$ | $10^{-4}$ | $10^{-4}$ | $10^{-4}$ |
| Label smoothing $\varepsilon$ | ✗ | ✗ | ✗ | ✗ | 0.2 |
| Entire scene as input | ✗ | ✗ | ✓ | ✗ | ✓ |
| Random rotation | ✗ | ✓ | ✗ | ✓ | ✓ |
| Random scaling | ✗ | [0.8,1.2] | [0.9,1.1] | ✗ | [0.9,1.1] |
| Random translation | ✗ | ✗ | ✗ | ✗ | ✗ |
| Random jittering | ✗ | 0.001 | ✗ | ✗ | ✓ |
| Height appending | ✗ | ✓ | ✗ | ✗ | ✓ |
| Color drop | ✗ | 0.2 | ✗ | ✗ | 0.2 |
| Color auto-contrast | ✗ | ✗ | ✓ | ✗ | ✓ |
| Color jittering | ✗ | ✗ | ✓ | ✗ | ✗ |
| mIoU (%) | 56.1 | 70.6 | 73.5 | 54.5 | 74.9 |

Table II: **Training strategies used in different methods for ScanObecjectNN classification.**

| Method | DGCNN | PointMLP | PointNet++ | PointNeXt (Ours) |
|---|---|---|---|---|
| Epochs | 250 | 200 | 250 | 250 |
| Batch size | 32 | 32 | 16 | 32 |
| Optimizer | Adam | SGD | Adam | AdamW |
| LR | $1 \times 10^{-3}$ | 0.01 | $10^{-3}$ | $2 \times 10^{-3}$ |
| LR decay | step | cosine | step | cosine |
| Weight decay | $10^{-4}$ | $10^{-4}$ | $10^{-4}$ | 0.05 |
| Label smoothing $\varepsilon$ | 0.2 | 0.2 | ✗ | 0.3 |
| Point resampling | ✗ | ✗ | ✗ | ✓ |
| Random rotation | ✓ | ✗ | ✓ | ✓ |
| Random scaling | ✗ | ✓ | ✗ | ✓ |
| Random translation | ✗ | ✓ | ✗ | ✗ |
| Random jittering | ✓ | ✗ | ✓ | ✗ |
| Height appending | ✗ | ✗ | ✗ | ✓ |
| OA (%) | 78.1 | 85.7 | 77.9 | 87.7 |

# E   Societal Impact

We do not see an immediate negative societal impact from our work. We notice that the way we discover the improved training and scaling strategies may consume a little more computing resources and affect the environment. Nevertheless, the improved training and scaling strategies will make researchers pay more attention to aspects other than architectural changes, which in the long term makes research in computer vision more diverse and generally better.

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

Table III: **Training strategies used in different methods for ScanNet segmentation.**

| Method | KPConv | PointTransformer | Stratified Transformer | PointNet++ | PointNeXt (Ours) |
|---|---|---|---|---|---|
| Epochs | 500 | 100 | 100 | 200 | 100 |
| Batch size | 10 | 16 | 8 | 32 | 2 |
| Optimizer | SGD | SGD | AdamW | Adam | AdamW |
| LR | $1 \times 10^{-2}$ | $5 \times 10^{-1}$ | $6 \times 10^{-3}$ | $1 \times 10^{-3}$ | $1 \times 10^{-3}$ |
| LR decay | step | multi step | multi step with warm up | step | multi step |
| Weight decay | $10^{-3}$ | $10^{-4}$ | $5 \times 10^{-2}$ | $10^{-4}$ | $10^{-4}$ |
| Entire scene as input | ✗ | ✓ | ✓ | ✗ | ✓ |
| Random rotation | ✓ | ✗ | ✓ | ✓ | ✓ |
| Random scaling | [0.9,1.1] | [0.9,1.1] | [0.8,1.2] | ✗ | [0.8,1.2] |
| Random translation | ✗ | ✗ | ✗ | ✗ | ✗ |
| Random jittering | 0.001 | ✗ | ✗ | ✗ | ✗ |
| Height appending | ✓ | ✗ | ✗ | ✗ | ✓ |
| Color drop | ✗ | ✗ | 0.2 | ✗ | 0.2 |
| Color auto-contrast | ✗ | ✓ | ✗ | ✗ | ✓ |
| Color jittering | ✗ | ✓ | ✗ | ✗ | ✗ |
| Test mIoU (%) | 68.6 | - | 73.7 | 55.7 | 71.2 |

Table IV: **Training strategies used in different methods for ShapeNetPart segmentation.**

| Method | DGCNN | KPConv | PointNet++ | PointNeXt (Ours) |
|---|---|---|---|---|
| Epochs | 201 | 500 | 201 | 300 |
| Batch size | 16 | 16 | 32 | 8 |
| Optimizer | Adam | SGD | Adam | AdamW |
| LR | $3 \times 10^{-3}$ | $1 \times 10^{-2}$ | $1 \times 10^{-3}$ | 0.001 |
| LR decay | step | step | step | multi step |
| Weight decay | 0.0 | $10^{-3}$ | 0.0 | $10^{-4}$ |
| Label smoothing $\varepsilon$ | ✗ | ✗ | ✗ | ✗ |
| Random rotation | ✗ | ✗ | ✗ | ✓ |
| Random scaling | ✗ | [0.9,1.1] | ✗ | [0.8,1.2] |
| Random translation | ✗ | ✗ | ✗ | ✗ |
| Random jittering | ✗ | 0.001 | ✓ | 0.001 |
| Normal Drop | ✗ | ✗ | ✗ | ✓ |
| Height appending | ✗ | ✓ | ✗ | ✓ |
| mIoU (%) | 85.2 | 86.4 | 85.1 | 87.0 |

[4] Xu Ma, Can Qin, Haoxuan You, Haoxi Ran, and Yun Fu. Rethinking network design and local geometry in point cloud: A simple residual MLP framework. In *International Conference on Learning Representations (ICLR)*, 2022.

[5] Charles Ruizhongtai Qi, Li Yi, Hao Su, and Leonidas J. Guibas. Pointnet++: Deep hierarchical feature learning on point sets in a metric space. In *Advances in Neural Information Processing Systems (NeurIPS)*, 2017.

[6] Hugues Thomas, Charles R Qi, Jean-Emmanuel Deschaud, Beatriz Marcotegui, François Goulette, and Leonidas J Guibas. Kpconv: Flexible and deformable convolution for point clouds. In *Proceedings of the IEEE/CVF International Conference on Computer Vision (ICCV)*, 2019.

[7] Mikaela Angelina Uy, Quang-Hieu Pham, Binh-Son Hua, Duc Thanh Nguyen, and Sai-Kit Yeung. Revisiting point cloud classification: A new benchmark dataset and classification model on real-world data. In *Proceedings of the IEEE/CVF International Conference on Computer Vision (ICCV)*, 2019.

[8] Yue Wang, Yongbin Sun, Ziwei Liu, Sanjay E. Sarma, Michael M. Bronstein, and Justin M. Solomon. Dynamic graph cnn for learning on point clouds. *ACM Transactions on Graphics (TOG)*, 2019.

[9] Li Yi, Vladimir G Kim, Duygu Ceylan, I Shen, Mengyan Yan, Hao Su, ARCewu Lu, Qixing Huang, Alla Sheffer, Leonidas Guibas, et al. A scalable active framework for region annotation in 3d shape collections. *ACM Transactions on Graphics (TOG)*, 35(6):210, 2016.

[10] Hengshuang Zhao, Li Jiang, Jiaya Jia, Philip HS Torr, and Vladlen Koltun. Point transformer. In *Proceedings of the IEEE/CVF International Conference on Computer Vision (ICCV)*, pages 16259–16268, 2021.

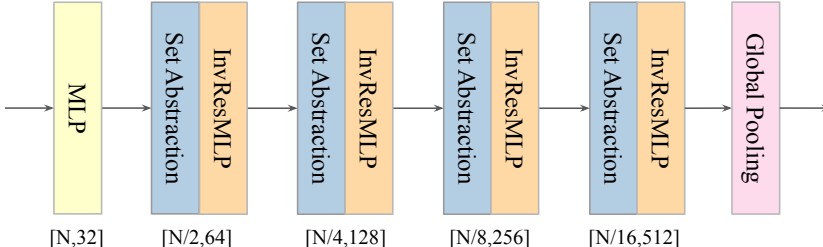

Figure I: **PointNeXt architecture for classification.** The classification architecture shares the same encoder as the segmentation architecture.

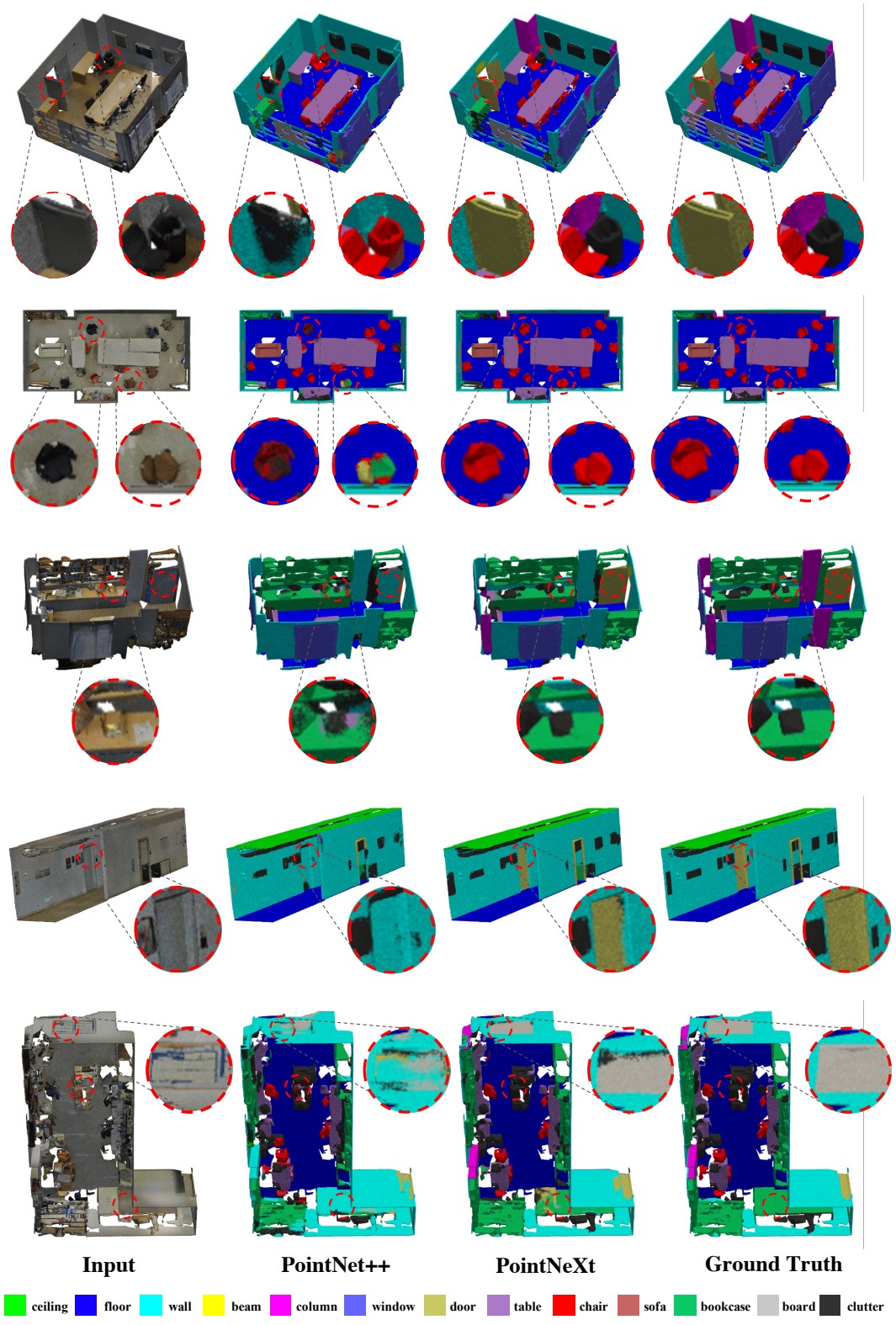

**Input**      **PointNet++**      **PointNeXt**      **Ground Truth**

🟩 ceiling 🟦 floor 🟦 wall 🟨 beam 🟪 column 🟦 window 🟨 door 🟪 table 🟥 chair 🟫 sofa 🟩 bookcase ⬜ board ⬛ clutter

Figure II: **Qualitative comparisons of PointNet++ ($2^{nd}$ column), PointNeXt ($3^{rd}$ column), and Ground Truth ($4^{th}$ column) on S3DIS semantic segmentation**. The input point cloud is visualized with original colors in the $1^{st}$ column. Differences between PointNet++ and PointNeXt are highlighted with red dash circles. Zoom-in for details.

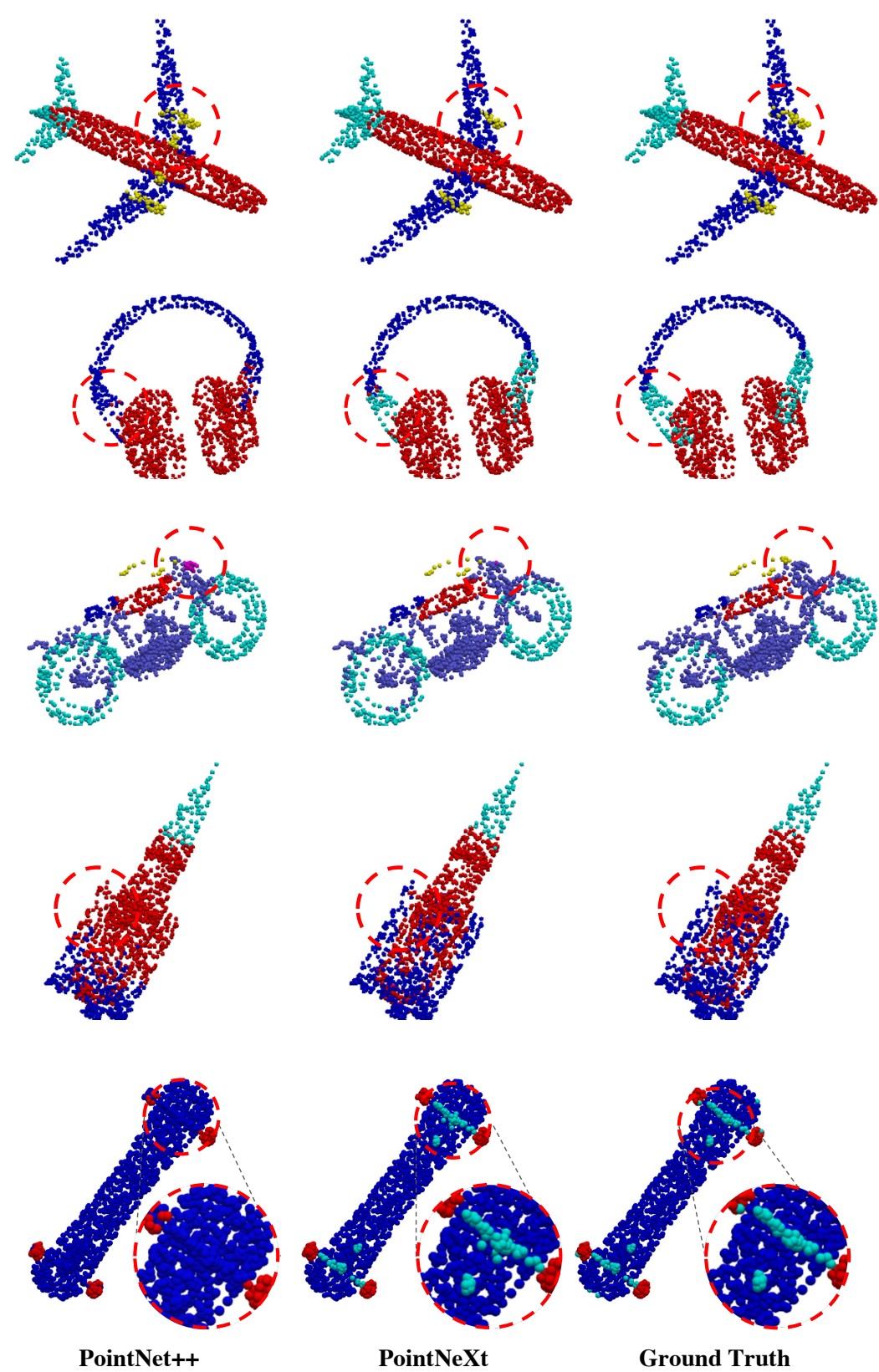

**PointNet++**  **PointNeXt**  **Ground Truth**

Figure III: **Qualitative comparisons of PointNet++ (left), PointNeXt (middle), and Ground Truth (right) on ShapeNetPart part segmentation**.