# OpenReview forum: "PointNeXt: Revisiting PointNet++ with Improved Training and Scaling Strategies"
_NeurIPS.cc/2022/Conference — NeurIPS 2022 Accept_

### Official Review · Reviewer_fSmm · 2022-07-10

**Rating:** 7
**Confidence:** 5
**Soundness:** 3 good
**Presentation:** 3 good
**Contribution:** 3 good

**Summary:**

In this paper, authors work on the point cloud understanding task and point out that we may need to revisit the training startegies and scaling methods in thi field. By detailed analysis and experiments, authors claimed that a large improvement in this field is introduced by these two aspects, rather than the network architecture designs in recent years. Furthermore, authors also introduce micro designs to improvement the PointNet++, and achieve state-of-the-art performance in three benchmarks.

I like this paper, revisting the training strategies and micro designs instead of proposing something "novel" and new. I will give an accpet.

**Questions:**

1. In L45， authors claimed that "For the S3DIS segmentation benchmark, can increase by 13.6% ". Is this simply achieved by training strategies (i.e., data augmentation and optimization techniques) and without any modifications to network structure?

2. I wonder why the authors consider the InvBlock design? What if we change it to a bottleneck structure in ResNet?

3. From Tab.4 and Tab. 5, we can see that normalizing the relative positions by Eq. (2) could only improve the performance by 0.3\% for both ScanObjectNN and S3DIS. Compared to other modifications, it could not be considered a "significant influence (L138)".

4. In L142, the authors claimed: "without normalization, ... larger weight..." I am interested in the distributions of position encoding layer weights. Also, if the authors cloud explain L143, "This makes ..., ... weight decay..." What results would weight decay cause?

**Limitations:**

Authors addressed the limitations and  potential negative societal impact of their work.

**Strengths And Weaknesses:**

Strengths:
1. I like the motivation in this paper. A revisiting in point cloud understanding is necessary and can help others greatly.
2. The detailed experiments and analysis make the experimental results solid.
3. The presentation and organization are good.

In general, I like this paper and recommend accept.

Weaknesses:
1. In L138, the authors claimed that "the radius is dataset-specific ..." This phenomenon is not a finding. We can always expect detailed hyper-parameters can cause varying performances on different datasets. A general fixed setting is appreciated.
2. InvResMLP present nothing new. the Inv-Res-Block has been proposed by MobileNetV2 and ConvNeXt already.
3. For the experiments, I acknowledge authors repeated all experiments three times. However, a critical issue is that all three datasets are small. To prove the main idea's effectiveness, I suggest authors conduct experiments on large datasets, like ScanNet for segmentation and SUN RGB-D for object detection.

---

> ### Author Response · Authors · 2022-08-02
> **Response to Reviewer #4 fSmm**
>
> Thank you for the positive recognition of our work and the valuable comments. We address the issues as follows:
>
> > **Q**: *"The radius is dataset-specific" is not a finding.*.
>
> **A**: Thank you for pointing this out. We will revise the paper to address this concern.
>
> > **Q**: *Inv-Res-Block presents nothing new*.
>
> **A**: To the best of our knowledge, we are the first to study such an inverted residual block in the point cloud field. This simple module is also practical for point cloud understanding and can help achieve state-of-the-art performance. We believe that this unified design between images and point clouds also benefits the computer vision community.
>
> > **Q**: *Conduct experiments on large datasets*.
>
> **A**: We conducted experiments on S3DIS, which is well-known for its large scale in the point cloud domain. One room in S3DIS dataset contains an average of 795K points, which is even over five times larger than ScanNet (around 146K points per room) in terms of the number of points.
>
> Here we provide additional experiments on ScanNet. We use similar training strategies on S3DIS for ScanNet experiments. Random rotation, random scaling, color auto-contrast, random color dropping, and height appending are used as data augmentations. Models are trained with CrossEntropy loss, AdamW optimizer, an initial learning rate of 0.001, a multistep learning rate scheduler (decay rate 0.1 at epochs 70 and 90), and a batch size of 16 for 100 epochs. The initial radius is set to 0.05. Results of PointNet++ and PointNeXt optimized with the above training strategies on the ScanNet *Val set* are provided in Table R4-A. PointNet++ trained with our strategies achieves 57.2 mIoU, outperforming the originally reported values by 3.7 mIoU. Our PointNeXt-B outperforms the PointNet++ by 14.5 mIoU. Our largest variant PointNeXt-XL achieves **72.5** mIoU, outperforming the voxel-based method MinkowskiNet. Note we do NOT perform any voting. Due to the limited time, we can only train and evaluate PointNet++, PointNeXt-B, and PointNeXt-XL.
>
> Table R4-A: Scene segmentation on Scannet Val Set.
> | Method | OA (%) | mAcc (%) |mIoU (%) |
> | -------- | -------- | -------- |-------- |
> | Point Transformer    | -|-	|70.6|
> | MinkowskiNet    | -  | -	|72.0| 1.0 |
> | Stratified Transformer    | -|-	|74.3|
> | PointNet++    | -|-	|53.5|
> | **PointNet++ (Ours)**  | 83.0	|67.5|	57.2|
> | **PointNeXt-B (Ours)**  | 88.5	|76.0|	68.0|
> | **PointNeXt-XL (Ours)**   | 89.8	| 79.6|	72.5|
>
> > **Q**: *In Line 45, "For the S3DIS segmentation benchmark, can increase by 13.6% ". Is it achieved by training strategies*.
>
> **A**: Yes, this is the result of PointNet++ with our training strategies alone without any change to the architecture.
>
> > **Q**: *Why consider InvBlock*.
>
> **A**: This design is inspired by Transformer i.e., where the hidden dimension of the MLP block is four times wider than the input dimension. InvBlock is also a successful module for image understanding proposed in MobileNetV2.
>
> > **Q**: *What if change to bottleneck structure*.
>
> **A**: We thank the reviewer for this question.  We replace the inverted structure with a bottleneck structure (bottleneck ratio = 4) in PointNeXt-XL. As shown in Table R4-B, the inverted block outperforms the bottleneck counterpart.
>
>
> Table R4-B: Scene segmentation on S3DIS.
> | Method | OA (%) | mAcc (%) | mIoU (%) |
> | -------- | -------- | -------- |-------- |
> | PointNeXt-XL with Bottleneck Structure    | $89.6\pm0.4$|	$75.3\pm0.2$	| $69.1\pm0.2$|
> | PointNeXt-XL   | **$90.6\pm0.2$**	|**$76.8\pm0.7$**|	**$70.5\pm0.3$**|
>
>
> > **Q**: *Compared to other modifications, normalizing the relative positions by Eq. (2) could not be considered a "significant influence (Line 138)".*
>
> **A**: We thank the reviewer for pointing this out. We will revise the paper to tone down this claim.

---

> > ### Author Response · Authors · 2022-08-02
> > **Response to Reviewer #4 fSmm. Relative position normalization.**
> >
> >
> > > **Q**: *In Line 142, "without normalization, ... larger weight...", distributions of position encoding layer weights*.
> >
> > **A**: We investigate the distributions of weights of the last position encoding layer in the pretrained PointNeXt-XL with and without relative coordinate normalization. Interestingly, we find PointNeXt-XL with normalization learns position weights  (mean -0.06, std 1.1) close to the weights of features (mean 0.05, std 1.8) in terms of magnitude. However,   PointNeXt-XL without normalization only learns position weights  (mean -0.007, std 0.5) 10 times smaller than features weights (mean -0.05, std 0.2) in terms of magnitude. This shows that without normalization, the network tends to be optimized to ignore relative positions because of their small values.
> >
> > > **Q**: *In Line 143, "This makes ..., ... weight decay..." What results would weight decay cause?*
> >
> > **A**: Weight Decay is a regularization technique that tends to reduce the magnitude of the weights towards zero. The relative position is small without normalization, so the Neural Network needs larger weights to apply $\Delta p$, otherwise, the effect of the relative position will be neglectable. This increases optimization difficulties.

---

> > > ### Comment · Reviewer_fSmm · 2022-08-05
> > > **Thanks for authors response and addtional experiments**
> > >
> > >  After reading other reviewers' comments and authors' rebuttals, I would keep my score and suggest acceptance.
> > >
> > > As I stated previously, one reason I like this work is that it provides systematical analysis and comparisons between recent works instead of proposing something new but trivial.  I do think this paper may inspire the community and help a lot.
> > >
> > > However, the contributions in architecture (say, Inv-Res-Block, weight decay) still cannot convince me.
> > >
> > > Also, for the PointNeXt-XL with Bottleneck Structure, authors should keep the parameters number and FLOPs same as PointNeXt-XL for a fair comparison (by adding blocks), not simply replacing to bottleneck structure.

---

> > > > ### Author Response · Authors · 2022-08-08
> > > > **Thanks for reviewer's recognition and encouraging comments**
> > > >
> > > >
> > > > Thank you for the recognition of our work. We appreciate your encouraging comments: "it provides systematical analysis and comparisons" and "may inspire the community and help a lot," which are exactly the original intention of our study. We address your additional concerns as follows.
> > > >
> > > >
> > > > > **Q**: *the contributions in architecture (say, Inv-Res-Block, weight decay) still cannot convince me.*
> > > >
> > > >
> > > > **A**: (i) With regards to Inv-Res-Block, PointNeXt is the first to show that InvResMLP inspired by 2D architectures can work effectively for 3D point cloud understanding. By combining InveResMLP and PointNet++, we propose a new architecture that achieves SOTA performance. (ii) With regards to relative position normalization, we find that without it, the network is learned to ignore the relative positions since their values are of small magnitude.
> > > >
> > > >
> > > > > **Q**: *PointNeXt-XL with Bottleneck Structure*
> > > >
> > > >
> > > >  We thank the reviewer for this suggestion. Here, we provide four additional experiments. (1) PointNeXt-XL with Bottleneck (C=64 -> C=65) to make the throughput the same as PointNeXt-XL by width scaling (increasing channels); (2) PointNeXt-XL with Bottleneck (C=64 -> C=70) to make the FLOPs the same as  PointNeXt-XL by width scaling; (3) PointNeXt (C=140) to make the number of parameters the same as PointNeXt-XL by width scaling; (4) PointNeXt-XL with Bottleneck (B=[3,6,3,3] -> B=[4,7,4,4]) to make the number of parameters the same as PointNeXt-XL by depth scaling (appending more blocks). PointNeXt-XL outperforms all the variants with bottleneck structures.
> > > >
> > > > **Table R4-B: Ablation study on bottleneck structures on S3DIS.**
> > > > | Method                                     | mIoU (%)     | Params (M) | FLOPs (G) | Throughput |
> > > > |--------------------------------------------|--------------|------------|-----------|------------|
> > > > | PointNeXt-XL   | $70.5\pm0.3$ |      41.5     |    84.8  |    45
> > > > | PointNeXt-XL with Bottleneck (Baseline)    | $69.1\pm0.2$ |       8.7     |      71.0     |    50         |
> > > > | PointNeXt-XL with Bottleneck (C=65)        | $67.9$            |    9.0        |        73.2   |     45       |
> > > > | PointNeXt-XL with Bottleneck (C=70)        |  	$68.9$           |  10.4          |     84.8      |     39       |
> > > > | PointNeXt-XL with Bottleneck (C=140)       |    $68.3$          |     41.6       |     336.3      |     18       |
> > > > | PointNeXt-XL with Bottleneck (B=[4,7,4,4]) |   $69.8$           |   10.8         |     87.2      |    43        |

---

> > > > > ### Comment · Reviewer_fSmm · 2022-08-08
> > > > > **Thanks for further responses**
> > > > >
> > > > > Thanks for the further responses.
> > > > >
> > > > > 1. Author claimed "PointNeXt is the first to show that ..."
> > > > > I would kind remind the authors that "first to introduce something from B to A" does not mean it is good or novel.
> > > > >
> > > > > 2.  Thanks for the further experiments, that makes more sense. And I believe fine-tunning the hyper-parameters for bootleneck structure carefully could achieve bettter performance.

---

> > > > > > ### Author Response · Authors · 2022-08-08
> > > > > > **Response to the further comments**
> > > > > >
> > > > > > Thank you for the constructive comments.
> > > > > >
> > > > > > > Q: *I would kind remind the authors that "first to introduce something from B to A" does not mean it is good or novel.*
> > > > > >
> > > > > > **A:** We thank the reviewer for pointing this out. We highlight that our main contributions are the systematical analysis of the training strategies in various benchmarks and the finding that the performance of the classical network PointNet++ can be improved to the state-of-the-art level. Apart from the main contributions, our introduced InvResMLP module is also proven to be effective. We thank that our contributions are recognized by you and other reviewers.
> > > > > >
> > > > > >
> > > > > > > Q: *I believe fine-tunning the hyper-parameters for bootleneck structure carefully could achieve bettter performance.*
> > > > > >
> > > > > > **A:** We thank the reviewer for this constructive suggestion. We will further explore the bottleneck structure and add this study in our revision.

---

### Official Review · Reviewer_jBW5 · 2022-07-10

**Rating:** 4
**Confidence:** 5
**Soundness:** 3 good
**Presentation:** 3 good
**Contribution:** 2 fair

**Summary:**

This paper explores a variety of tricks to improve the performance of PointNet++ for point cloud classification and segmentation. They first show that by incorporating modern training strategies and data augmentations, the vanilla version of PointNet++ can be improved significantly. Then, to scale the PointNet++ for better performance, they propose the InvResMLP that adopts residual connection and separate MLPs in Set Abstraction layer to support scalable PointNet++, where the relative coordinates are also normalized to ease the model optimization. The experiments demonstrate that the proposed improved version of PointNet++ (PointNeXt) achieves promising performance on 3D semantic segmentation (S3DIS), 3D object classification (ScanObjectNN) and 3D part segmentation (ShapeNetPart).

**Questions:**

How about the performance of PointNeXt in larger datasets like ScanNet? Can it be extended to larger outdoor scenes like SemanticKITTI?

**Limitations:**

The limitations have been well discussed in the paper.

**Strengths And Weaknesses:**

Strengths:
* The paper conducts extensive experiments to explore various data augmentation and training strategies, which greatly improve the performance of vanilla PointNet++.
* Although simple, the proposed normalization of relative coordinate is reasonable for easing the optimization.
* The experiments show that the proposed InvResMLP can enable a larger model to improve the model performance.

Weaknesses:
*  Although the reviewer agrees that the authors have systematically explored the potential of vanilla PointNet++, the technical contribution of this paper is still not enough to be published at NeurIPS: see below comments.
* The explored data augmentation and training strategies are well-known tricks for training a point cloud model, and it is also well-known that vanilla PointNet++ can achieve much better performance than the original officially reported performance.
* The proposed InvResMLPis is also straightforward by adding a residual connection (to solve vanishing gradient) and moving more MLPs after max-pooling (to reduce calculations).
* The PointNet++ is hard to scalable to larger point cloud scenes, such as some outdoor scenes, and also the performance of PointNet++ series is inferior to sparse-voxel-based networks (like in larger dataset ScanNet). This paper does not show that an improved version of PointNet++ can still be the next-generation framework for point cloud processing in such a larger dataset.

---

> ### Author Response · Authors · 2022-08-02
> **Response to Reviewer #3 jBW5**
>
> We respect the reviewer's opinion, however, we believe that our work strongly alines with NeurIPS standards and has a positive impact on the community. We address the reviewer's comments below:
>
> > **Q**: *data augmentation and optimization strategies are well-known tricks*.
>
> **A**: We agree it is known that applying more advanced training strategies mostly leads to better performance. However, our work is the first to formally and comprehensively study the incremental effect of those strategies. Specifically, our work answers the following two *previously unsolved* fundamental questions, (i) *which* training strategies to apply, and (ii) *how much improvement* can the training strategies lead to. The merits of our work include that, first, we *quantify the effect of each strategy* and provide a state-of-the-art combination of training strategies for each benchmark. Second, we showcase that simply by adopting our training strategies, without any change to the architecture, the performance of PointNet++ can be improved significantly (see Tab. 4 and Tab. 5 for specific numbers of improvements), and even surpass SOTA in some cases (Tab. 4).
>
>
> > **Q**: *InvResMLP is straightforward*.
>
> **A**: Our work does not aim to propose any new architectural innovations. Instead, we choose to look in an orthogonal equally important direction which is network training strategies. With regards to the architecture modification, we borrow and combine successful architectural designs from the 2D domain (as cited in our paper) and introduce them to the field of point clouds (the proposed InvResMLP module). Surprisingly, we are the first to show that this simple InvResMLP module can work effectively in 3D point cloud understanding and allow us to outperform the SOTA. We believe that this finding is of great benefit to the community.
>
>
> > **Q**: *hard to scale to large point cloud scenes*. Can PointNeXt be extended to larger scenes like ScanNet and SemanticKITTI?
>
> **A**: We appreciate the reviewer's comment. We argue that PointNeXt can be easily deployed to any large-scale dataset. In the manuscript, we have shown the superiority of PointNeXt in a large-scale dataset, Stanford *Large-Scale* 3D Indoor Spaces dataset (S3DIS). Note that S3DIS is well-known for its large scale in the point cloud domain, where one room contains an average of 795K points, which is even over five times larger than ScanNet (around 146K points per room) in terms of the number of points.
>
>
> Nevertheless, we do understand the encouragement for the additional experiments on ScanNet and SemanticKIITI. Due to the limited rebuttal time, we choose to focus on ScanNet. We use similar training strategies on S3DIS. Random rotation, random scaling, color auto-contrast, random color dropping, and height appending are used as data augmentations. Models are trained with CrossEntropy loss, AdamW optimizer, an initial learning rate of 0.001, a multistep learning rate scheduler (decay rate 0.1 at epochs 70 and 90), and a batch size of 16 for 100 epochs. The initial radius is set to 0.05. Results of PointNet++ and PointNeXt optimized with the above training strategies on the ScanNet *Val set* are provided in Table R3-A. PointNet++ trained with our strategies achieves 57.2 mIoU, outperforming the originally reported values by 3.7 mIoU. Our PointNeXt-B outperforms the PointNet++ by 14.5 mIoU. The largest variant PointNeXt-XL achieves 72.5 mIoU, outperforming the voxel-based method MinkowskiNet. Note we do NOT perform any voting. Due to the limited time, we can only train and evaluate PointNet++, PointNeXt-B, and PointNeXt-XL. Our experiments show that PointNeXt can be extended to ScanNet and achieves better performance than the point-based methods Point Transformer and the voxel-based method MinkowskiNet.
>
>
> Table R3-A: Scene segmentation on ScanNet Val Set.
> | Method | OA (%) | mAcc (%) |mIoU (%) |
> | -------- | -------- | -------- |-------- |
> | Point Transformer    | -|-	|70.6|
> | MinkowskiNet    | -  | -	|72.0| 1.0 |
> | Stratified Transformer    | -|-	|74.3|
> | PointNet++    | -|-	|53.5|
> | **PointNet++ (Ours)**  | 83.0	|67.5|	57.2|
> | **PointNeXt-B (Ours)**  | 88.5	|76.0|	68.0|
> | **PointNeXt-XL (Ours)**   | 89.8	| 79.6|	72.5|
>
> Results of previous methods in Table R3-A are from Stratified Transformer [1].
>
> [1] Lai, Xin, Jianhui Liu, Li Jiang, Liwei Wang, Hengshuang Zhao, Shu Liu, Xiaojuan Qi, and Jiaya Jia. "Stratified Transformer for 3D Point Cloud Segmentation." In Proceedings of the IEEE/CVF Conference on Computer Vision and Pattern Recognition, pp. 8500-8509. 2022.

---

> > ### Comment · Reviewer_jBW5 · 2022-08-04
> > **Response to Authors' Rebuttal**
> >
> > Thanks for providing the rebuttal.
> >
> > I agree that this paper has a positive impact to the community by systematically investigating the data augmentation and PointNet++ to achieve better performance.
> >
> > However, the technical contribution and the generalizability of PointNet++ are still the main concern, and this paper is more like a meaningful technical report but not a good academic publication. From the reviewer's perspective, the only new thing is the normalization of relative coordinates, but the experiments show that it has slight performance gains (also mentioned by other reviewers).
> >
> >
> > For the rebuttal, I have the following questions:
> >
> > * Large-scale dataset: the scale of the dataset not only contains the number of points but also the perception region.
> >     * Although S3DIS contains a huge number of points, the region is very small (indoor scene), and very dense points within a small region don't make sense since you can easily voxelize the points and only segment the sampled points and extend to neighboring points.
> >     * In contrast, the point cloud in outdoor scenes makes more sense due to a real point cloud and much larger perception region. Are PointNet++ still applicable to achieve good performance on such a large perception region?  Compared with the voxel-based method, can it still achieve a good trade-off between performance and efficiency? Considering the limited time, the reviewer does not really need very good performance but need the authors to convince the reviewer that it is feasible.
> >
> > * For the experiments on ScanNet, can you also report the performance on test set leaderboard?  Since this paper tailors to a high-performance framework with sorts of engineering techniques, the comparison with state-of-the-art works on test set is important.
> >
> > * Since the paper argues that this is the first comprehensive study about  data augmentation and optimization strategies, can you compare the differences with PointTransformer in terms of these techniques? It would convince the reviewer more that this work is meaningful instead of writing something other paper adopted but not mentioned in their paper.

---

> > > ### Author Response · Authors · 2022-08-07
> > > **Response to Reviewer's discussion part 1**
> > >
> > > Thank you for the constructive suggestions. We address the concerns below:
> > >
> > >
> > > > **Q**: *Concerns about technical contribution.*
> > >
> > > **A**: We respect the reviewer's opinion. With regards to technical contributions, we would like to highlight two points. 1) Our work is a comprehensively empirical study, presenting a surprising finding (especially in the current landscape of increasingly sophisticated and complicated deep networks): with improved training and model scaling strategies, the performance of the simple baseline PointNet++ can be significantly improved and outperform SOTA. We think that this finding would bring some meaningful impact to the community and facilitate future innovations along this path. 2) PointNeXt is the first to show that InvResMLP inspired by 2D architectures can work effectively for 3D point cloud understanding. By combining InveResMLP and PointNet++, we propose a new architecture that gets SOTA performance. This successful introduction of existing modules from one domain to another domain is also commonly deemed novel in the literature.
> > >
> > >
> > > > **Q**: *Performance of PointNet++ in outdoor scenes with large reception region.*.
> > >
> > > **A**: Thank you for pointing out this new scenario. PointNeXt is applicable in outdoor scenes and can achieve reasonable performance with real-time efficiency. In Table R3-C, we show that the reported performance of PointNet++ (20.1 mIoU) can be increased by **+28.4** mIoU to reach 48.4 mIoU in the SemanticKITTI test set while being 4 times faster, by adopting our proposed training strategies, a dataset-specific radius (0.4m), relative position normalization, and the random sampling in replacement of the original furthest point sampling. Below we share the technical details in SemanticKITTI.
> > >
> > >
> > > There are two significant differences between segmenting outdoor scenes in SemanticKIITI and segmenting indoor scenes in ScanNet / S3DIS: (i) the perception region is much larger (~100 meters v.s. ~10 meters), and (2) high efficiency requirement. Due to (i), we ablate the *initial radius and find that simply increasing it from 0.1m to 0.4m gives around 10 mIoU higher in validation*. Because of (ii) we replace furthest point sampling in PointNet++ with *random sampling* as suggested by RandLA-Net [1]. Random rotation, random scaling, random jittering, and height appending are used as data augmentations. Models are trained with CrossEntropy loss, AdamW optimizer, an initial learning rate of 0.001, a multistep learning rate scheduler (decay rate 0.1 at epochs 30 and 40), and a batch size of 16 for 50 epochs.
> > >
> > >
> > > **Table R3-C: Ourdoor semantic segmentation on SemanticKITTI.**
> > >
> > > | Method                        | val mIoU | test mIoU | Throughput |
> > > | ----------------------------- | -------- | --------- | ---------- |
> > > | PointNet                      | -        | 14.6      |      -      |
> > > | RandLA-Net                    |      -    | 53.9      |       -     |
> > > | PointNet++                    | -        | 20.1      |    10        |
> > > | **PointNet++ (Ours, r=0.1m)** | 37.3     | 35.4 (+15.3) |      42      |
> > > | **PointNet++ (Ours, r=0.4m)** | 47.2     |  -       |       42   |
> > > | **PointNeXt-S**               | 48.9   | 48.4 (+28.3) |       42     |
> > >
> > >
> > >
> > >
> > > [1] Hu, Qingyong, Bo Yang, Linhai Xie, Stefano Rosa, Yulan Guo, Zhihua Wang, Niki Trigoni, and Andrew Markham. "Randla-net: Efficient semantic segmentation of large-scale point clouds." In Proceedings of the IEEE/CVF Conference on Computer Vision and Pattern Recognition, pp. 11108-11117. 2020.
> > >
> > >
> > > > **Q**: *Results on ScanNet Test Set*.
> > >
> > > We provide the results on ScanNet Test Set in Table R3-D. Per submission policy of ScanNet, one paper can only submit one result per track per two weeks [2]. We are only able to get the testing result of PointNeXt-XL at this time. PointNeXt achieves 72.5 mIoU in validation and 71.5 mIoU in testing. Note that we do not perform voting in validation and testing unlike previous methods (e.g. Stratified Transformer). Experiments show that PointNeXt can be extended to ScanNet and is able to increase the originally reported performance of PointNet++ by **+15.5 mIoU** by the improved training and model scaling strategies.
> > >
> > > **Table R3-D: Scene segmentation on ScanNet.**
> > > | Method                  | val mIoU (%) | test mIoU (%) |
> > > | ----------------------- | ------------ | ------------- |
> > > | Point Transformer       | 70.6         | -             |
> > > | MinkowskiNet            | 72.0         | 73.4          |
> > > | Stratified Transformer  | 74.3         | 73.7          |
> > > | PointNet++              | 53.5         | 55.7          |
> > > | **PointNet++ (Ours)**   | 57.2 (+3.7)        | -             |
> > > | **PointNeXt-B (Ours)**  | 68.0 (+14.5)       | -             |
> > > | **PointNeXt-XL (Ours)** | 72.5 (+19)        | 71.2 (+15.5)          |
> > >
> > >
> > > [2] https://kaldir.vc.in.tum.de/scannet_benchmark/documentation#submission-policy

---

> > > > ### Author Response · Authors · 2022-08-07
> > > > **Response to Reviewer's discussion part 2**
> > > >
> > > >
> > > > > **Q**: *Differences with PointTransformer in terms of training strategies?*
> > > >
> > > > We have provided the training differences between our PointNeXt and Point Transformer in S3DIS in the submitted **supplementary material** (Table IV). For easy review, we additionally show the comparison of training strategies in Table R3-E. PointNeXt differs from Point Transformer in the optimizer, the LR scheduler, label smoothing, random rotation, random jittering, height appending, color dropping, and color jittering. Pls refer to Table R3-E for details.
> > > >
> > > > Note that Point Transformer did not officially conduct experiments in ScanNet. The reported values of Point Transformer in Table R3-D are from Stratified Transformer [3]. We compare the training strategies between our PointNeXt and Stratified Transformer in ScanNet in Table R3-F. PointNeXt differs from Stratified Transformer in the lr, LR decay, weight decay, random jittering, and color auto-contrast.  Pls see Table R3-F for details.
> > > >
> > > > [3] Lai, Xin, Jianhui Liu, Li Jiang, Liwei Wang, Hengshuang Zhao, Shu Liu, Xiaojuan Qi, and Jiaya Jia. "Stratified Transformer for 3D Point Cloud Segmentation." In Proceedings of the IEEE/CVF Conference on Computer Vision and Pattern Recognition, pp. 8500-8509. 2022.
> > > >
> > > > **Table R3-E: Training strategies used in PointTransformer and our PointNeXt in S3DIS**
> > > > | $\textbf{Method}$               | $\textbf{PointTransformer}$ | $\textbf{PointNeXt (Ours)}$ |
> > > > |-------------------------------|:-------------------------:|:-------------------------:|
> > > > | Epochs                        |            100            |            100            |
> > > > | Batch size                    |             16            |             32            |
> > > > | Optimizer                     |            SGD            |           AdamW           |
> > > > | LR                            |            0.5            |           $0.01$          |
> > > > | LR decay                      |         multi step        |           cosine          |
> > > > | Weight decay                  |         $10^{-4}$         |         $10^{-4}$         |
> > > > | Label smoothing $\varepsilon$ |          0.0          |            0.2            |
> > > > | Random rotation               |          no          |          yes         |
> > > > | Random scaling                |           [0.9, 1.1]               | [0.9, 1.1]                          |   |
> > > > | Random jittering              |          no          |           sigma=0.005         |
> > > > | Height appending              |          no          |           yes          |
> > > > | Color dropping                |          no       |            0.2            |
> > > > | Color auto-contrast           |          yes          |         yes         |
> > > > | Color jittering               |          yes        |          no          |
> > > > | mIoU (\%)                     |            73.5           |            74.9           |
> > > >
> > > > **Table R3-F: Training strategies used in StratifiedTransformer and our PointNeXt in ScanNet**
> > > > | $\textbf{Method}$             | $\textbf{StratifiedTransformer}$ | $\textbf{PointNeXt (Ours)}$ |
> > > > |-------------------------------|----------------------------------|-----------------------------|
> > > > | Epochs                        | 100                              | 100                         |
> > > > | Batch size                    | 8                                | 8                           |
> > > > | Optimizer                     | AdamW                            | AdamW                       |
> > > > | LR                            | 0.006                            | $0.001$                     |
> > > > | LR decay                      | multi step with warm up          | multi step                  |
> > > > | Weight decay                  | 0.05               | $10^{-4}$                   |
> > > > | Label smoothing $\varepsilon$ | 0.0                              | 0.0                         |
> > > > | Random rotation               | yes                              | yes                         |
> > > > | Random scaling                | [0.8, 1.2]                       | [0.8, 1.2]                  |
> > > > | Random jittering              | no                               | sigma=0.005                 |
> > > > | Height appending              | no                               | yes                         |
> > > > | Color dropping                | 0.2                              | 0.2                         |
> > > > | Color auto-contrast           | no                               | yes                         |
> > > > | Color jittering               | no                               | no                          |
> > > > | Test mIoU (\%)                | 73.7                             | 71.2                          |

---

> > > > > ### Comment · Reviewer_jBW5 · 2022-08-08
> > > > > **Thanks for the feedback**
> > > > >
> > > > > Thanks for the detailed feedback!
> > > > >
> > > > > There are some further questions for discussion:
> > > > > * Can you comment a bit about why PointNeXt is still applicable and suitable to large-scale outdoor scenes? Since its performance (48+%) is much lower than SOTA (60-70% mIoU), and is also lower than other Point-based method like KPConv (58.8%) and RandLA-Net (53.9%).
> > > > > * How many points does PointNeXt utilize in each level?  Do you apply 0.4 radius in each level? Is it enough to capture a large scene with a small number of points (e.g., 256) and 0.4 radius?
> > > > > * What's the advantage of PointNeXt compared with voxel-based method? Especially for the large-scale outdoor scenes. Speed or performance? This is the most important concern. A good illustration and analysis can convince the reviewer that the improvement on PointNet++ is meaningful to the whole 3D community. And the reviewer does not understand why PointNeXt can be faster than voxel-based methods since there are lots of customized CUDA operations (FPS, set abstraction).

---

> > > > > > ### Author Response · Authors · 2022-08-08
> > > > > > **Response to Reviewers' further questions**
> > > > > >
> > > > > > We thank the reviewer for the additional comments.
> > > > > >
> > > > > > > Q: *why PointNeXt is still applicable and suitable for large-scale outdoor scenes?*
> > > > > >
> > > > > > **A:** We successfully show that the performance of PointNet++ can be surprisingly **improved by over 20 mIoU** by **only** adopting improved training and model scaling strategies. We highlight that this achievement is made **within only 4 days**. Given more parameter tuning (e.g. radius, learning rate, data augmentations), we believe the mIoU can be further improved to achieve at least a close performance to other point-based methods.
> > > > > >
> > > > > > However, we do acknowledge that compared to voxel-based methods, the point-based methods (not only PointNeXt) fail to prove their strength for outdoor scenes in the current landscape. We think this inferiority is because of the non-uniform nature of outdoor scenes, which causes difficulties for neighbor querying and local aggregation in point-based methods. We will mention this in the limitation part and leave it as future work.
> > > > > >
> > > > > >
> > > > > > > Q: *How many points does PointNeXt utilize in each level?*
> > > > > >
> > > > > > **A:** We use an *initial* radius of 0.4m. The radius is doubled by default when the point cloud is downsampled (mentioned in manuscript L136-137). Since we have four stages in the PointNeXt architecture, the radius for the last stage is 6.4m, which can cover core parts of most objects of interest, such as cars, pedestrians, e.t.c. We note that there might be a radius other than 0.4m that can lead to better performance. With regards to the number of points, we always query k=32 neighbors, which is the same as PointNet++.
> > > > > >
> > > > > > > Q: *What's the advantage of PointNeXt compared with the voxel-based method? Especially for the large-scale outdoor scenes.*
> > > > > >
> > > > > > **A:** View-based, voxel-based, and point-based are the three mainstream point cloud processing schemes. All of them are widely used. Point-based methods are comparable to the voxel-based methods in indoor scene perception, and especially dominate the application where the input point cloud is small-scale. The focus of our work is the point-based method, where we show the classical point-based method PointNet++ can be improved to reach SOTA point-based performance.  Nevertheless, we agree with the reviewer that the point-based methods fail to prove their strength for outdoor scenes in the current landscape (please refer to question 1 for details).
> > > > > >
> > > > > >
> > > > > > > Q: *Why PointNeXt can be faster than voxel-based methods since there are lots of customized CUDA operations (FPS, set abstraction)?*
> > > > > >
> > > > > > **A:** We mentioned in the discussion part I that we replaced FPS with random sampling to speed up PointNet++ by 4 times from 10 ins./sec. to 42 ins./sec.

---

> > > > > > > ### Comment · Reviewer_jBW5 · 2022-08-08
> > > > > > > **Thanks for the feedback**
> > > > > > >
> > > > > > > Thanks very much for the additional experiments and comments!
> > > > > > >
> > > > > > > The reviewer does acknowledge the experiments on a new dataset SemanticKITTI within a very short time:
> > > > > > > * I agree with you that the improved training strategies + larger radius can achieve better performance, which is also well-known for lots of researchers in the 3D community.
> > > > > > > * The inferior performance does verify the concern of the reviewer that the structure of PointNet++ is not suitable for large-scale scenes, actually we also do not notice that there are any other recent approaches using this PointNet++-like structure as a backbone network.
> > > > > > >
> > > > > > > Nevertheless, I have to say that the most concerns are still the technical contributions of this paper:
> > > > > > > * The exhaustive ablation experiments about some well-known data augmentation.
> > > > > > > * A residual block
> > > > > > >
> > > > > > > Actually, these two contributions can be applied to lots of structures, not just PointNet++. Although the reviewer acknowledges the efforts and experiments of the authors to show a clear engineering exploration of these factors, both of them are not significant to be published at NeurIPS.
> > > > > > >
> > > > > > > As for the surprising finding that PointNet++ can also achieve very good performance on multiple datasets, actually, it is not that surprising since both the reviewer and some other researchers have known this fact for a long time, maybe just because the baseline of PointNet++ is not that strong.
> > > > > > >
> > > > > > > According to the above reasons, I can not recommend an accept rating.

---

> > > > > > > > ### Author Response · Authors · 2022-08-09
> > > > > > > > **Thank you for the comments. We highlight few things here for people interested in our work.**
> > > > > > > >
> > > > > > > >
> > > > > > > > Thank you for the comments and the request for additional experiments. We appreciate the time spent by the reviewer. We would like to highlight the following:
> > > > > > > >
> > > > > > > >
> > > > > > > > 1. Our paper does not try to push the rope towards neither point-based methods nor voxel-based methods. Our work focuses on point-based methods and aims at improving the classical PointNet++ without sophisticated architecture engineering.
> > > > > > > >
> > > > > > > >
> > > > > > > > 2. New observations and findings are also meaningful and helpful for the community. The intention of this work is not to propose new architecture, yet trying to draw the attention of the point cloud community to the training and scaling strategies, which are truly important. Although it is known by *some* people that improving training strategies can lead to *good* performance for PointNet++, it is **NOT known** by *many*: (i) **what** are the better training strategies, and (ii) a particular **finding** that the performance of PointNet++ can be raised to reach or even outperform **SOTA** point-based methods.

---

### Official Review · Reviewer_8bUj · 2022-07-11

**Rating:** 6
**Confidence:** 4
**Soundness:** 3 good
**Presentation:** 3 good
**Contribution:** 3 good

**Summary:**

The paper aims to improve PointNet++ by slightly modifying PointNet++, increasing model scale and employing a few training strategies. Comprehensive experiments on S3DIS for semantic segmentation, ScanObjectNN for real-world object classification, and ShapeNetPart for object part segmentation.

**Questions:**

1. The novelty seems a bit limited. The receptive field scaling (normalization) looks interesting and technically sound. However, according to the ablation study, the improvement is not that significant.

2. In Table 4 and Table 5, it cloud be better to include SGD as a reference.


**Ethics Review Area:**

["I don’t know"]

**Limitations:**

No significant limitations.

**Strengths And Weaknesses:**

1. Although the novelty seems limited, the paper may have a positive effect on the community. Some existing works (even actually also based on PointNet++) just follow the trend and apply popular techniques to point clouds without very convincing reasons. The paper may cool down such behaviours.

2. The paper looks solid. The proposed method achieves several SOTAs on multiple tasks and datasets.

---

> ### Author Response · Authors · 2022-07-30
> **Response to reviewer 8bUj**
>
> We thank the reviewer for the constructive suggestions and valuable feedback. We answer all the comments below and will revise the manuscript accordingly.
>
> > **Q**: *Relative coordinates normalization is not significant*.
>
> **A**: Thank you for the interest in our relative coordinates normalization. Note that we did not claim normalization can significantly improve performance. The benefit of the proposed normalization is that it *improves performance consistently without an overhead*. It is a single-line code that averagely produces +0.4 mIoU on S3DIS (Tab. 5) and +0.3 OA on ScanObjectNN (Tab. 4). We also found that normalization has a larger impact on bigger models (e.g., +2.3 mIoU for PointNeXt-XL, Tab. 7) compared to smaller models (e.g., +0.4 mIoU for PointNet++). This might be because the bigger model is harder to optimize compared to the smaller models, and our normalization allows for achieving an easier optimization.
>
> > **Q**: *Include SGD as a reference*.
>
> **A**: We thank the reviewer for this helpful suggestion. We provide the results of SGD here and will add them to the revised manuscript. Note that SGD might require a different learning rate (lr) compared to Adam and AdamW, thus we test SGD with different learning rates (lr * 0.1, lr, lr * 10, lr * 50, lr * 100). In the tables below, we include the results of PointNet++ optimized using SGD with the best learning rate on ScanObjectNN and S3DIS, respectively. AdamW produces better results than SGD and Adam on both benchmarks.
>
>
> | ScanObjectNN                            |   OA (\%)    | $\Delta$ |
> | --------------------------------------- | :----------: | :------: |
> | PointNet++                              |     77.9     |    --    |
> | ...                                     |     ...      |   ...    |
> | $+$ Label Smoothing                     | $85.0\pm0.5$ |   +1.3   |
> | $+$ Adam $\rightarrow$ AdamW (lr=0.002) | $85.6\pm0.1$ |   +0.6   |
> | $+$ AdamW $\rightarrow$ SGD (lr=0.002)  | $84.8\pm0.1$ |   -0.8   |
>
>
>
> | S3DIS                                  |   mIoU (%)   | $\Delta$ |
> | -------------------------------------- | :----------: | :------: |
> | PointNet++                             |     51.5     |    --    |
> | ...                                    |     ...      |   ...    |
> | $+$ Label Smoothing                    | $61.9\pm0.1$ |   +0.4   |
> | $+$ Adam $\rightarrow$ AdamW (lr=0.01) | $62.5\pm0.6$ |   +0.6   |
> | $+$ AdamW $\rightarrow$ SGD (lr=0.5)   | $59.4\pm0.5$ |   -3.1   |

---

> > ### Comment · Reviewer_8bUj · 2022-08-08
> > **Responses to Authors' Rebuttal**
> >
> > Thank you for the responses, which address my concern regarding the effectiveness of the relative coordinate normalization.
> >
> > I recognize the contribution of the paper but its novelty is not that significant. Therefore, I lean towards keeping my rating 6.

---

### Official Review · Reviewer_LDZj · 2022-07-12

**Rating:** 5
**Confidence:** 5
**Soundness:** 3 good
**Presentation:** 3 good
**Contribution:** 3 good

**Summary:**

As the title suggests, the paper revisits and studies PointNet++, a popular neural network for point cloud representation learning. Based on the study, the paper discovers two results: first, it proposes a set of training strategies to boost the performance on PointNet++ so that it achieves SOTA results on various benchmarks; and second, it shows how PointNet++ can be modified and scaled to further boost performance.

**Questions:**

Several questions have been asked in the Weakness sections. Overall the paper has merits but there are parts where the paper is weak. I will update my review based on the rebuttal.

**Strengths And Weaknesses:**

Strengths:
- The paper achieves impressive performance on various benchmarks (87.7% on ScanObjectNN, 74.9% mean IOU on S3IS, 87.2 IOU on ShapeNetPart)
- The paper conducts a systematic study showing the improvement in performance because of various factors (Table 4 and Table 5). In my opinion, such findings are useful.
- The paper adopts ways to improve the PoinNet++  architecture. This could be useful for future research as they can use a more improved and modern version of PointNet++.
- The paper shows how the findings about training strategies are generalizable to network architectures in Table 6, however, this result should be made more rigorous as discussed in the weaknesses section.

Weaknesses:
- The study showing that the findings about training strategies are generalizable to other networks is very useful (as discussed in the strengths) but limited. Specifically, only two other networks have been evaluated. I would suggest evaluating at least one simple model like SimpleView and a recent model like PointMLP.
- It is unclear why information about FLOPs has been omitted in Table 2 and Table 3. It would be important for readers to know the strengths and limitations of various methods. Similarly, information about Parameter Count is only omitted in Table 3 for the ShapeNetPart dataset. It is unclear why specific parameters are omitted for different datasets in the main paper.
- In Table 2, PointNet, DGCNN, and additional models (SimpleView, PointMLP) with improved training strategy should be added, so that readers can quantify the difference in performance purely as a result of network architecture and also compare models across parameter count, throughput and FLOPs.
- Potential misrepresentation of prior work: The paper says “However, SimpleView simply adopts the same training strategies as DGCNN. In contrast, we conduct a systematic study to quantify the effect of each data augmentation and optimization technique”. It seems like a misrepresentation of prior work as SimpleView does compare models across different augmentation and optimization schemes, and finds that the ones used in DGCNN are better. The paper should provide more information about what is meant in this comparison.
- Information about scaling is confusing. From my understanding, PointNeXt-S is the smaller valiant (L182-L185), however, in Table 4 PointNeXt-S is mentioned alongside Scale-up. Similarly, scaling rules for ShapeNetPart are not consistent (Table 3). Does the usual scaled-up model not perform well on this dataset? If so, this should be discussed.

---

> ### Author Response · Authors · 2022-07-30
> **To reviewer #1 LDZj**
>
> Thank you for the constructive suggestions and valuable feedback. All comments are addressed below. We will revise the paper in light of the provided comments.
>
> > **Q**: *Generalization of the training strategies*.
>
> **A**: Along with PointNet++, we provided three other networks optimized with our training strategies, including *PointMLP*, PointNet, and DGCNN in Tab. 6. The proposed training and scaling strategies can improve the overall accuracy (OA) of PointMLP by 1.7 from 85.4±0.3 to 87.1±0.7 on ScanObjectNN. The results and analysis of PointMLP have been discussed in the manuscript (Line 323-330).
>
> As suggested, we further evaluate our training strategies on SimpleView [1]. Note that SimpleView is a view-based method, whereas our training strategies focused on point-based methods. Therefore, some of the proposed strategies might not be applicable for view-based methods, e.g., height appending. Despite that, we can still observe that our training strategies improve the performance of SimpleView by 1.5 from 80.5 to 82.0±0.4 OA. We will report this result with relevant discussion in the revision.
>
> [1] Goyal, Ankit, et al. "Revisiting point cloud shape classification with a simple and effective baseline." International Conference on Machine Learning. PMLR, 2021.
>
> > **Q**: *Missing parameters or FLOPs*.
>
> **A**: Model parameters and FLOPs were omitted in Tab. 2 and Tab. 3 because of the limited space in the manuscript. Here we provide the number of parameters, FLOPs, and throughputs of representative methods on ScanObjectNN in Tab. R1-A and on ShapeNetPart in Tab. R1-B. The results in the tables show that PointNeXt surpasses state-of-the-art point-based methods with fewer parameters and lower computational costs. We will provide the number of parameters, FLOPs, and throughput in the revision.
>
>
> Table R1-A: Object classification on ScanObjectNN.
> | Method                      | OA                                     | mAcc                                    | Params. (M) | FLOPs (G)|   Throughput (ins./sec.)  |
> |---------------------------------------|----------------------------------------|-----------------------------------------|:-------:|:-----:|:-------------:|
> | DGCNN          | 78.1                                   | 73.6                                    |   1.8   |  4.8  |      402      |
> | DGCNN (**Ours**)         |            $\textbf{86.0}\pm0.5$                         | $\textbf{84.0}\pm0.7$                                        |   1.8   |  4.8  |      402      |
> | PointMLP       | $85.4\pm1.3$                           | $83.9\pm1.5$                            |   13.2  |  31.4 |      191      |
> | PointMLP (**Ours**)       | $\textbf{87.1}\pm0.7$                           | $\textbf{85.6}\pm1.1$                          |   13.2  |  31.4 |      191      |
> | SimpleView | 80.5                                        |   -                                |   0.8   |    6.7   |    102           |
> | SimpleView (**Ours**) | $\textbf{82.0}\pm0.4$                                        |   $\textbf{78.4}\pm0.5$                                |   0.8   |    6.7   |    102        |
> | PointNet++ | 77.9                                   | 75.4                                    |   1.5   |  1.7  |      1872     |
> | PointNet++ (**Ours**) | $86.1\pm0.7$	| $84.2\pm0.9$                                                   |   1.5   |  1.7  |      1872    |
> | PointNeXt-S (**Ours**)       | $\textbf{87.7}\pm0.4$ | $\textbf{85.8}\pm0.6$|   1.4   |  1.6  |      2040     |
>
>
>
> Table R1-B: Part segmentation on ShapeNetPart.
> | Method              | Ins. mIoU      | Params. (M)| FLOPs (G)| Throughput (ins./sec.) |
> |---------------------|----------------|:-------:|:-----:|:-------------:|
> | PointNet            | 83.7           | 3.6     | 4.9   | 1184       |
> | DGCNN               | 85.2           | 1.3     | 12.4  | 147        |
> | CurveNet            | 86.8           | 5.5       | 5.1    | 97         |
> | PointNet++          | 85.1           | 1.0     | 4.9   | 705        |
> | PointNeXt-S         | $86.5 \pm 0.0$   | 1.0     | 4.5   | 782        |
> | PointNeXt-S (C=64)  | $86.9 \pm 0.1$ | 3.7     | 17.6  | 431        |
> | PointNeXt-S (C=160) | $87.2 \pm 0.0$ | 22.3    | 109.3 | 77         |
>
> > **Q**: *Add results with improved training strategies in Tab. 2*.
>
> **A**: We thank the reviewer for this helpful suggestion. We will add them into Tab. 2 in the revision.
>
> > **Q**: *Potential misrepresentation of prior work*.
>
> **A**: Thank you for pointing this out. We will revise the paper to reflect that SimpleView compares models across different training strategies. Moreover, we will highlight that the performance of SimpleView can be further improved using our training strategies.

---

> > ### Author Response · Authors · 2022-08-02
> > **response to Review #1's comment on scaling information**
> >
> > > **Q**: *Information about scaling is confusing*.
> >
> > **A**: (i) As mentioned in Line 167-169, PointNet++ used different model configurations in classification, part segmentation, and semantic segmentation tasks. Compared to PointNet++ for classification and part segmentation in terms of the number of convolutional layers, PointNeXt-S is a scaled-up variant. Compared to PointNet++ for semantic segmentation on S3DIS, PointNeXt-S is a scaled-down variant. We will revise the paper to make this point more clear.
> >
> > (ii) We thank the reviewer for pointing out that the scaling rule for ShapeNetPart differs from others. We found that the performance was saturated with depth scaling on ScanObjectNN and ShapeNetPart. This is mainly due to the small scales of these two datasets (refer to Line 255-264). We will clarify this in the revision.

---

### Author Response · Authors · 2022-08-09
**General Responses to  Reviewers and ACs**

Dear reviewers and ACs:

We sincerely thank all reviewers for their insightful feedback and constructive suggestions.


We would like to emphasize that our work not only introduces a simple yet effective module InvResMLP for scaling up PointNet++. Our work also proposes a systematical analysis of the modern training strategies in the point cloud domain. Although it is known by some people that improving training strategies can lead to good performance, it is not known by many until our work: (i) *what* are the better training strategies, and (ii) *how much improvement exactly can each strategy lead to*. We show that, without any architecture change, vanilla PointNet++ optimized with the improved training strategies can achieve comparable performance to SOTA. By further scaling up PointNet++, its performance can be raised to outperform SOTA point-based methods in various benchmarks (S3DIS, ScanObjectNN, and ShapeNetPart). We believe these new observations and findings are rather meaningful and helpful for the community.


With regards to the revision, we will incorporate the insightful suggestions of the reviewers as follows:

According to Reviewer 1's suggestions, we will revise Tab. 3 to add results of representative models optimized with the improved training strategies.

According to Reviewer 2’s suggestions, we will add SGD baselines in the ablation study.

According to the suggestions from Reviewer 3 and Reviewer 4, we will add the experiment in the larger-scene dataset ScanNet in revision. We find PointNet++ with improved training and scaling strategies can reach a comparable performance to MinkowskiNet and Point Transformer.

According to Reviewer 3's comments, we will also mention the limitation of the current point-based methods in outdoor scene understanding, and provide our findings in SemanticKITTI in the supplementary.

According to Reviewer 4’s suggestions, we will further explore the bottleneck structure and add this study in revision.

We will improve other minor points mentioned by all reviewers in revision.

Thank you all for the valuable suggestions.

Thanks,

Paper 703 Authors

---

> ### Public Comment · ~Guocheng_Qian1 · 2023-01-03
> **Camera Ready Version Done**
>
> We revise the camera ready version accordingly. Note the results of representative models optimized with the improved training strategies are included only in Tab. 6 instead of Tab. 3 to save space. We thank all the reviewers for their suggestions.

---

### Meta-Review · Area_Chair_rksy · 2022-08-25

**Recommendation:** Accept
**Confidence:** Less certain

**Metareview:**

This paper presents a series of training strategies and settings that can improve PointNet++ to match the performance of state-of-the-art architectures. The AC agrees with reviewer jBW5 that the novelty of the paper is limited and some phenomena were observed before. However, the detailed training strategies might have the potential to benefit the research community. Open-sourcing the code will be therefore important.



**Award:**

No

---

### Decision · Program_Chairs · 2022-09-14

Accept